# FINITE-TIME ANALYSIS OF SINGLE-TIMESCALE ACTOR-CRITIC ON LINEAR QUADRATIC REGULATOR

## ABSTRACT

Actor-critic (AC) methods have achieved state-of-the-art performance in many challenging tasks. However, their convergence in most practical applications are still poorly understood. Existing works mostly consider the uncommon double-loop or two-timescale stepsize variants for the ease of analysis. We investigate the practical yet more challenging single-sample single-timescale natural AC for solving the canonical linear quadratic regulator problem. Specifically, the actor and the critic update only once with a single sample in each iteration using proportional stepsizes. We prove that the single-sample single-timescale natural AC(NAC) can attain an $\epsilon$-optimal solution with a sample complexity of $\mathcal{O}(\epsilon^{-2})$, which elucidates on the practical efficiency of single-sample single-timescale NAC. We develop a novel analysis framework that directly bounds the whole interconnected iteration system without the conservative decoupling commonly adopted in previous analysis of AC and NAC. Our work presents the first finite-time analysis of single-sample single-timescale NAC with a global optimality guarantee.

## 1 INTRODUCTION

Actor-critic (AC) methods achieved substantial success in solving many difficult reinforcement learning (RL) problems (LeCun et al., 2015; Mnih et al., 2016; Silver et al., 2017). In addition to a policy update, AC methods employ a parallel critic update to bootstrap the Q-value for policy gradient estimation, which often enjoys reduced variance and fast convergence in training.

Despite the empirical success, theoretical analysis of AC in the most practical form remains challenging. Most existing works focus on either the double-loop setting or the two-timescale setting, both of which are uncommon in practical implementations. In double-loop AC, the actor is updated in the outer loop only after the critic takes sufficiently many steps to have an accurate estimation of the Q-value in the inner loop (Yang et al., 2019; Kumar et al., 2019; Wang et al., 2019). Hence, the convergence of critic is decoupled from that of the actor. The analysis is separated into a policy evaluation sub-problem in the inner loop and a perturbed gradient descent in the outer loop. In two-timescale AC, the actor and the critic are updated simultaneously in each iteration using stepsizes of different timescales. The actor stepsize (denotes by $\alpha_t$) is typically smaller than that of the critic (denotes by $\beta_t$), with their ratio goes to zero as the iteration number goes to infinity (i.e., $\lim_{t\to\infty} \alpha_t/\beta_t = 0$). The two-timescale allows the critic to approximate the correct Q-value in an asymptotic way. This design essentially decouples the analysis of the actor and the critic.

The aforementioned AC variants are considered mainly for the ease of analysis. In practice, the single-timescale AC, where the actor and the critic are updated simultaneously using constantly proportional stepsizes (i.e., with $\alpha_t/\beta_t = c_\alpha > 0$), is more favorable due to its simplicity of implementation and empirical sample efficiency (Schulman et al., 2015; Mnih et al., 2016). However, its analysis is significantly more difficult than the other variants. To understand its finite-time convergence, some recent works (Fu et al., 2020; Zhou & Lu, 2022) consider multi-sample variants of single-timescale AC, where the critics are updated by the least square temporal difference (LSTD) estimator rather than the TD(0) update. The idea is still to obtain an accurate policy gradient estimation at each iteration by using sufficient samples (LSTD), and then follows the common perturbed gradient analysis to guarantee the convergence of the actor, decoupling the convergence analysis of the actor and the critic. In addition to the multi-sample settings, there are few attempts that analyzed the single-sample single-timescale AC(NAC), and they only attest local convergence (Chen et al.,

2021; Olshevsky & Gharesifard, 2022). Besides, (Olshevsky & Gharesifard, 2022) only considers the simple tabular case. We attempt to answer the more general yet more challenging question:

*Can the single-sample single-timescale AC(NAC) find a global optimal policy, especially on the general unbounded continuous state-action space with unbounded reward?*

To this end, we make the first step to consider the classic Linear Quadratic Regulation (LQR), a fundamental continuous state-action space control problem that are commonly employed to study the performance and the limits of RL algorithms (Fazel et al., 2018; Yang et al., 2019; Tu & Recht, 2018; Duan et al., 2021). In particular, under the time-average cost, the single-sample single-timescale AC(NAC) algorithm for solving LQR consists of three parallel updates in each iteration: the cost estimator, the critic, and the actor. Unlike the aforementioned double-loop, two-timescale, or multi-sample structures, there is no specialized design in single-sample single-timescale AC(NAC) that facilitates a decoupled analysis of its three interconnected updates. In fact, it is both conservative and difficult to bound the three iterations separately. Moreover, the existing perturbed gradient analysis can no longer be applied to establish the convergence of the actor either. To tackle these challenges in analysis, we instead propose a novel framework to directly bound the overall interconnected iteration system altogether, without resorting to conservative decoupled analysis. In particular, despite the inaccurate estimation in all three updates, we prove the estimation errors diminish to zero if the (constant) ratio of the stepsizes between the actor and the critic is below a threshold. The identified threshold provides new insights into the practical choices of the stepsizes for single-timescale AC.

Overall, our contributions are summarized as follows:

• Our work furthers the theoretical understanding of AC(NAC) in its most practical form. We for the first time show that the single-sample single-timescale NAC can provably find the $\epsilon$-accurate global optimum with a sample complexity of $\mathcal{O}(\epsilon^{-2})$ for tasks with unbounded continuous state-action space. The previous works consider either specialized algorithm variants (Fu et al., 2020; Zhou & Lu, 2022), or more restricted settings with only local convergence guarantee (Chen et al., 2021; Olshevsky & Gharesifard, 2022).

• We also contribute to the work of RL on continuous control task. It is novel that even with actor updated by a roughly estimated gradient, the single-sample single-timescale NAC algorithm can still find the global optimal policy for LQR, under general assumptions. Compared with all other model-free RL algorithms for solving LQR (see related work 1.1), our work is the first one adopting the simplest single-sample single-loop structure, which may serve as the first step towards understanding the limits of AC(NAC) methods on continuous control task. In addition, compared with the state-of-the-art double-loop AC for solving LQR (Yang et al., 2019), we improve the sample complexity from $\mathcal{O}(\epsilon^{-5})$ to $\mathcal{O}(\epsilon^{-2})$. We also show the algorithm is much more sample-efficient empirically compared to a few classic works in Section 5, which unveils the practical wisdom of AC(NAC) algorithm.

• Technically, we provide a new proof framework that can establish the finite-time convergence for single-timescale AC. In the finite-time analysis of double-loop AC (Yang et al., 2019) and two-timescale AC (Wu et al., 2020), the previous techniques hinge on decoupling the analysis of actor and critic, establishing the convergence of critic first and then the convergence of actor consequently. The novelty of our proof framework is that we formulate the estimation errors of the time-average cost, the critic, and the natural policy gradient into an interconnected iteration system and establish the convergence for them simultaneously rather than separately. This proof framework may provide new insights for finite-time analysis of other single-timescale algorithms.

## 1.1 RELATED WORK

In this section, we review the existing works that are most relevant to ours.

**Actor-Critic methods.** The first AC algorithm was proposed by Konda & Tsitsiklis (1999). Kakade (2001) extended it to the natural AC algorithm. The asymptotic convergence of AC algorithms has been well established in Kakade (2001); Bhatnagar et al. (2009); Castro & Meir (2010); Zhang et al. (2020). Many recent works focused on the finite-time convergence of AC methods. Under the double-loop setting, Yang et al. (2019) established the global convergence of AC methods for solving LQR. Wang et al. (2019) studied the global convergence of AC methods with both the actor and the critic being parameterized by neural networks. Kumar et al. (2019) studied the finite-time

local convergence of a few AC variants with linear function approximation. Under the two-timescale AC setting, Wu et al. (2020); Xu et al. (2020) established the finite-time convergence to a stationary point at a sample complexity of $\mathcal{O}(\epsilon^{-2.5})$. Under the single-timescale setting, all the related works (Fu et al., 2020; Chen et al., 2021; Zhou & Lu, 2022; Olshevsky & Gharesifard, 2022) have been reviewed in the Introduction.

**RL algorithms for LQR.** RL algorithms in the context of LQR have seen increased interest in the recent years. These works can be mainly divided into two categories: model-based methods (Dean et al., 2018; Mania et al., 2019; Cohen et al., 2019; Dean et al., 2020) and model-free methods. Our main interest lies in the model-free methods. Notably, Fazel et al. (2018) established the first global convergence result for LQR under the policy gradient method using zeroth-order optimization. Krauth et al. (2019) studied the convergence and sample complexity of the LSTD policy iteration method under the LQR setting. On the subject of adopting AC to solve LQR, Yang et al. (2019) provided the first finite-time analysis with convergence guarantee and sample complexity under the double-loop setting. Zhou & Lu (2022) considered the multi-sample (LSTD) and single-timescale setting. For the more practical yet challenging single-sample single-timescale AC, there is no such theoretical guarantee so far, which is the focus of this paper.

**Notation.** Without other specification, for two sequences $\{x_n\}$ and $\{y_n\}$, we write $x_n = \mathcal{O}(y_n)$ if there exists an constant $C$ such that $x_n \leq C y_n$. We use $\tilde{\mathcal{O}}(\cdot)$ to further hide logarithm factors. For any symmetric matrix $M \in \mathbb{R}^{n \times n}$, let $\text{svec}(M) \in \mathbb{R}^{n(n+1)/2}$ denote the vectorization of the upper triangular part of $M$ and $\text{smat}(\cdot)$ denote its inverse such that $\text{smat}(\text{svec}(M)) = M$. Finally, we denote by $A \otimes_s B$ the symmetric Kronecker product of two matrices $A$ and $B$.

## 2 PRELIMINARIES

In this section, we introduce the AC algorithm and provide the theoretical background of LQR.

### 2.1 ACTOR-CRITIC ALGORITHMS

We consider the reinforcement learning for the standard Markov Decision Process (MDP) defined by $(\mathcal{X}, \mathcal{U}, \mathcal{P}, c)$, where $\mathcal{X}$ is the state space, $\mathcal{U}$ is the action space, $\mathcal{P}(x_{t+1}|x_t, u_t)$ denotes the transition kernel that the agent transits to state $x_{t+1}$ after taking action $u_t$ at current state $x_t$, and $c(x_t, u_t)$ is the running cost. A policy $\pi_\theta(u|x)$ parameterized by $\theta$ is defined as a mapping from a given state to a probability distribution over actions.

In this paper, we aim to find a policy $\pi_\theta$ that minimizes the infinite-horizon time-average cost, which is given by

$$\theta^* = \arg\min_\theta J(\theta) := \lim_{T \to \infty} \mathbb{E}_\theta \frac{\sum_{t=0}^{T} c(x_t, u_t)}{T} = \mathbb{E}_{x \sim \rho_\theta, u \sim \pi_\theta} [c(x, u)], \quad (1)$$

where $\rho_\theta$ denotes the stationary state distribution generated by policy $\pi_\theta$. In the time-average cost setting, the state-action value (Q-value) of policy $\pi_\theta$ is defined as

$$Q_\theta(x, u) = \mathbb{E}_\theta[\sum_{t=0}^{\infty} (c(x_t, u_t) - J(\theta))|x_0 = x, u_0 = u], \quad (2)$$

which describes the accumulated differences between running costs and average cost for selecting $u$ in state $x$ and thereafter following policy $\pi_\theta$ (Sutton & Barto, 2018). Based on this definition, we can use the policy gradient theorem (Sutton et al., 1999) to express the gradient of $J(\theta)$ with respect to $\theta$ as

$$\nabla_\theta J(\theta) = \mathbb{E}_{x \sim \rho_\theta, u \sim \pi_\theta} [\nabla_\theta \log \pi_\theta(u|x) Q_\theta(x, u)].$$

One can also choose to update the policy using the natural policy gradient (Kakade, 2001), which is given by

$$\nabla_\theta^N J(\theta) = F(\theta)^\dagger \nabla_\theta J(\theta). \quad (3)$$

where

$$F(\theta) = \mathbb{E}_{x \sim \rho_\theta, u \sim \pi_\theta} [\nabla_\theta \log \pi_\theta(u|x) \nabla_\theta \log \pi_\theta(u|x)^\top]$$

is the Fisher information matrix and $F(\theta)^\dagger$ denotes its Moore Penrose pseudoinverse.

Optimizing $J(\theta)$ in (1) with (3) requires evaluating the Q-value of the current policy $\pi_\theta$, which is usually unknown. AC estimates both the Q-value and the policy. The critic update approximates Q-value towards the actual value of the current policy $\pi_\theta$ using temporal difference (TD) learning (Sutton & Barto, 2018). The actor improves the policy to reduce the time-average cost $J(\theta)$ via gradient descent. Note that the AC with natural policy gradient is also known as natural AC.

## 2.2 Natural Actor-Critic for Linear Quadratic Regulator

In this paper, we aim to demystify the convergence property of natural AC by focusing on the infinite-horizon time-average linear quadratic regulator (LQR) problem:

$$\underset{\{u_t\}}{\text{minimize}} \quad J(\{u_t\}) := \lim_{T \to \infty} \frac{1}{T} \mathbb{E}[\sum_{t=1}^{T} x_t^\top Q x_t + u_t^\top R u_t] \tag{4}$$

$$\text{subject to} \quad x_{t+1} = A x_t + B u_t + \epsilon_t, \tag{5}$$

where $x_t \in \mathbb{R}^d$ is a state and $u_t \in \mathbb{R}^k$ is a action; $A \in \mathbb{R}^{d \times d}$ and $B \in \mathbb{R}^{d \times k}$ are system matrices; $Q \in \mathbb{S}^{d \times d}$ and $R \in \mathbb{S}^{k \times k}$ are performance matrices; $\epsilon_t \sim \mathcal{N}(0, D_0)$ are i.i.d Gaussian random variables with $D_0 > 0$. From the optimal control theory (Anderson & Moore, 2007), the optimal policy of (4) is a linear feedback of the state

$$u_t = -K^* x_t, \tag{6}$$

where $K^* \in \mathbb{R}^{k \times d}$ is the optimal policy which can be uniquely found by solving an Algebraic Riccati Equation (ARE) (Anderson & Moore, 2007) depending on $A$, $B$, $Q$, $R$. This means that finding $K^\star$ using ARE relies on the complete model knowledge.

In the sequel, we pursue finding the optimal policy in a *model-free* way by using the natural AC method, without knowing or estimating $A, B, Q, R$. The structure of the optimal policy in (6) allows us to reformulate (4) as a static optimization problem over all feasible policy matrix $K \in \mathbb{R}^{k \times d}$. To encourage exploration, we parameterize the policy as

$$\{\pi_K(\cdot|x) = \mathcal{N}(-Kx, \sigma^2 I_k), K \in \mathbb{R}^{k \times d}\}, \tag{7}$$

where $\sigma > 0$ is the standard deviation of the exploration noise. In other words, given a state $x_t$, the agent will take an action $u_t$ according to $u_t = -K x_t + \sigma \zeta_t$, where $\zeta_t \sim \mathcal{N}(0, I_k)$. As a consequence, the closed-loop form of system (5) under policy (7) is given by

$$x_{t+1} = (A - BK)x_t + \xi_t, \tag{8}$$

where $\xi_t = \epsilon_t + \sigma B \zeta_t \sim \mathcal{N}(0, D_\sigma)$ with $D_\sigma = D_0 + \sigma^2 BB^\top$. Note that optimizing over the set of stochastic policies (7) will lead to the same optimal $K^*$.

The set $\mathbb{K}$ of all stabilizing policies is given by

$$\mathbb{K} := \left\{ K \in \mathbb{R}^{k \times d} : \rho(A - BK) < 1 \right\}, \tag{9}$$

where $\rho(\cdot)$ denotes the spectral radius. It is well known that if $K \in \mathbb{K}$, the Markov chain in (8) yields a stationary state distribution $\mathcal{N}(0, D_K)$, where $D_K$ satisfies the following Lyapunov equation

$$D_K = D_\sigma + (A - BK)D_K(A - BK)^\top. \tag{10}$$

Similarly, we define $P_K$ as the unique positive definite solution to

$$P_K = Q + K^\top R K + (A - BK)^\top P_K (A - BK). \tag{11}$$

Based on $D_K$ and $P_K$, the following lemma characterizes $J(K)$ and its gradient $\nabla_K J(K)$.

**Lemma 2.1.** *(Yang et al., 2019) For any $K \in \mathbb{K}$, the time-average cost $J(K)$ and its gradient $\nabla_K J(K)$ take the following forms*

$$J(K) = \text{Tr}(P_K D_\sigma) + \sigma^2 \text{Tr}(R), \tag{12a}$$

$$\nabla_K J(K) = 2 E_K D_K, \tag{12b}$$

*where $E_K := (R + B^\top P_K B)K - B^\top P_K A$.*

Then, the natural gradient of $J(K)$ can be calculated as (Fazel et al., 2018; Yang et al., 2019)

$$\nabla_K^N J(K) = \nabla_K J(K) D_K^{-1} = E_K, \tag{13}$$

which eliminates the burden of estimating $D_K$. Note that we omit the constant coefficient since it can be absorbed by the stepsize.

Calculating the natural gradient $\nabla_K^N J(K)$ requires estimating $P_K$, which depends on $A, B, Q, R$. To estimate the gradient without the knowledge of the model, we instead directly utilize the Q-value.

**Lemma 2.2.** *(Bradtke et al., 1994; Yang et al., 2019) For any $K \in \mathbb{K}$, the Q-value $Q_K(x, u)$ takes the following form*

$$Q_K(x, u) = (x^\top, u^\top) \Omega_K \begin{pmatrix} x \\ u \end{pmatrix} - \sigma^2 \mathrm{Tr}(R + P_K B B^\top) - \mathrm{Tr}(P_K D_K), \tag{14}$$

*where*

$$\Omega_K := \begin{bmatrix} \Omega_K^{11} & \Omega_K^{12} \\ \Omega_K^{21} & \Omega_K^{22} \end{bmatrix} := \begin{bmatrix} Q + A^\top P_K A & A^\top P_K B \\ B^\top P_K A & R + B^\top P_K B \end{bmatrix}. \tag{15}$$

Clearly, if we can estimate $\Omega_K$, then $E_k$ in (13) can be readily estimated by using $\Omega_K^{21}$ and $\Omega_K^{22}$.

## 3 SINGLE-SAMPLE SINGLE-TIMESCALE NATURAL ACTOR-CRITIC

In this section, we describe the single-sample single-timescale natural AC algorithm for solving LQR. In view of the structure of the Q-value given in (14), we define the following feature function

$$\phi(x, u) = \mathrm{svec}\left[\begin{pmatrix} x \\ u \end{pmatrix} \begin{pmatrix} x \\ u \end{pmatrix}^\top\right].$$

Then, we can parameterize the Q-estimator (critic) by

$$\hat{Q}_K(x, u; w, b) = \phi(x, u)^\top w + b.$$

Using the TD(0) learning, the critic update follows by

$$\omega_{t+1} = \omega_t + \beta_t[(c_t - J(K) + \phi(x_{t+1}, u_{t+1})^\top \omega_t + b - \phi(x_t, u_t)^\top \omega_t - b)]\phi(x_t, u_t), \tag{16}$$

where $\beta_t$ is the stepsize of the critic and $K$ denotes the policy under which the state-action pairs are sampled. Note that the constant $b$ is not required for updating the linear coefficient $\omega$.

Taking the expectation of $\omega_{t+1}$ in (16) with respect to the stationary distribution, conditioned on $\omega_t$, the expected subsequent critic can be written as

$$\mathbb{E}[\omega_{t+1}|\omega_t] = \omega_t + \beta_t(b_K - A_K \omega_t), \tag{17}$$

where

$$A_K = \mathbb{E}_{(x,u)}[\phi(x, u)(\phi(x, u) - \phi(x', u'))^\top], \quad b_K = \mathbb{E}_{(x,u)}[(c(x, u) - J(K))\phi(x, u)]. \tag{18}$$

Note that for ease of exposition, we denote $(x', u')$ as the next state-action pair after $(x, u)$ and abbreviate $\mathbb{E}_{x \sim \rho_K, u \sim \pi_K(\cdot|x)}$ as $\mathbb{E}_{(x,u)}$.

Given a policy $\pi_K$, it is not hard to show that if the update in (17) has converged to some limiting point $\omega_K^*$, i.e., $\lim_{t \to \infty} \omega_t = \omega_K^*$, $\omega_K^*$ must be the solution of $A_K \omega = b_K$.

**Proposition 3.1.** *Suppose $K \in \mathbb{K}$. Then the matrix $A_K$ defined in (18) is invertible and $A_K \omega = b_K$ has a unique solution $\omega_K^*$ that satisfies*

$$\omega_K^* = \mathrm{svec}(\Omega_K). \tag{19}$$

*where $\Omega_K$ is defined in (15).*

Combining (13), (15), and (19), we can express the natural gradient of $J(K)$ using $\omega_K^*$:

$$\nabla_K^N J(K) = \Omega_K^{22} K - \Omega_K^{21} = \text{smat}(\omega_K^*)^{22} K - \text{smat}(\omega_K^*)^{21}.$$

This allows us to estimate the natural policy gradient using the critic parameters $\omega_t$, and then update the actor in a model-free manner

$$K_{t+1} = K_t - \alpha_t \widehat{\nabla_{K_t}^N J}(K_t), \tag{20}$$

where $\alpha_t$ is the actor stepsize and $\widehat{\nabla_{K_t}^N J}(K_t)$ is the natural gradient estimation depending on $\omega_t$:

$$\widehat{\nabla_{K_t}^N J}(K_t) = \text{smat}(\omega_t)^{22} K_t - \text{smat}(\omega_t)^{21}. \tag{21}$$

Furthermore, we introduce a cost estimator $\eta_t$ to estimate the time-average cost $J(K_t)$. Combining the critic update (16) and the actor update (20), the single-sample single-timescale natural AC for solving LQR is listed below.

---

**Algorithm 1** Single-Sample Single-timescale Natural Actor-Critic for Linear Quadratic Regulator

---

1: **Input** initialize actor parameter $K_0 \in \mathbb{K}$, critic parameter $\omega_0$, time-average cost $\eta_0$, stepsizes $\alpha_t$ for actor, $\beta_t$ for critic, and $\gamma_t$ for cost estimator.
2: **for** $t = 0, 1, 2, \cdots, T-1$ **do**
3:      Sample $x_t$ from the stationary distribution $\rho_{K_t}$.
4:      Take action $u_t \sim \pi_{K_t}(\cdot|x_t)$ and receive $c_t = c(x_t, u_t)$ and the next state $x_t'$.
5:      Obtain $u_t' \sim \pi_{K_t}(\cdot|x_t')$.
6:      TD error calculation: $\delta_t = c_t - \eta_t + \phi(x_t', u_t')^\top \omega_t - \phi(x_t, u_t)^\top \omega_t$
7:      Cost estimator update: $\eta_{t+1} = \eta_t + \gamma_t(c_t - \eta_t)$
8:      Critic update: $\omega_{t+1} = \Pi_{\bar{\omega}}(\omega_t + \beta_t \delta_t \phi(x_t, u_t))$
9:      Actor update: $K_{t+1} = K_t - \alpha_t(\text{smat}(\omega_t)^{22} K_t - \text{smat}(\omega_t)^{21})$
10: **end for**

---

Note that "single-sample" refers to the fact that only one sample is used to update the critic per actor step. Line 3 of Algorithm 1 samples from the stationary distribution induced by the policy $\pi_{K_t}$, which is a mild requirement in the analysis of uniformly ergodic Markov chain, such as in the LQR problem (Yang et al., 2019). It is only made to simplify the theoretical analysis. Indeed, as shown in Tu & Recht (2018), when $K \in \mathbb{K}$, (8) is geometrically $\beta$-mixing and thus its distribution converges to the stationary distribution exponentially. In practice, one can run the Markov chain in (8) a sufficient number of steps and sample one state from the last step. In addition, "single-timescale" refers to the fact that the stepsizes for the critic and the actor updates are constantly proportional.

Since the update of the critic parameter in (16) requires the time-average cost $J(K_t)$, Line 7 provides an estimation of it. Besides, on top of (16), we additionally introduce a projection ($\Pi_{\bar{\omega}}$) in Line 8 to keep the critic norm-bounded, which is common in the literature (Wu et al., 2020; Yang et al., 2019; Xu et al., 2020). In our analysis, the projection is relaxed using its nonexpansive property.

## 4   MAIN THEORY

In this section, we establish the global convergence and analyze the finite-time performance of Algorithm 1. All the proofs can be found in the Appendix A.

Before preceding, we make the following standard assumptions.

**Assumption 4.1.** *There exists a constant $\bar{K} > 0$ such that $\|K_t\| \leq \bar{K}$ for all $t$.*

The above assumes the uniform boundedness of the actor parameter (Konda & Tsitsiklis, 1999; Karmakar & Bhatnagar, 2018; Barakat et al., 2022; Zhou & Lu, 2022). As can be seen from our proof, it is only made to guarantee the boundedness of the feature functions, which is a standard assumption in the literature of analyzing AC with linear function approximation (Xu et al., 2020; Wu et al., 2020; Zhou & Lu, 2022).

**Assumption 4.2.** *There exists a constant $\rho \in (0, 1)$ such that $\rho(A - BK_t) \leq \rho$ for all $t$.*

Assumption 4.2 is made to ensure the stability of the closed loop systems induced in each iteration and thus ensure the existence of the stationary distribution corresponding to policy $\pi_{K_t}$. In the single-sample case, the estimation of the natural gradient of $J(K_t)$ can be highly noisy and biased. In general, it is difficult to obtain a theoretical guarantee for this condition. Nevertheless, we will present numerical examples to support this assumption. Moreover, the assumption for the existence of stationary distribution is common and has been widely used in Chen et al. (2021); Zhou & Lu (2022); Olshevsky & Gharesifard (2022).

Under these two assumptions, we can now prove the convergence of Algorithm 1, which consists of three estimators: $\eta_t, \omega_t$, and $K_t$.

**Theorem 4.3.** *Suppose that Assumptions 4.1 and 4.2 hold and choose $\alpha_t = \frac{c_\alpha}{\sqrt{1+t}}, \beta_t = \gamma_t = \frac{1}{\sqrt{1+t}}$, where $c_\alpha$ is a small positive constant. With probability at least $1 - 10^{-10}$, we have*

$$\frac{1}{T} \sum_{t=0}^{T-1} \mathbb{E}(\eta_t - J(K_t))^2 = \mathcal{O}(\frac{1}{\sqrt{T}}),$$

$$\frac{1}{T} \sum_{t=0}^{T-1} \mathbb{E}\|\omega_t - \omega_{K_t}^*\|^2 = \mathcal{O}(\frac{1}{\sqrt{T}}),$$

$$\min_{0 \le t < T} \mathbb{E}[J(K_t) - J(K^*)] = \mathcal{O}(\frac{1}{\sqrt{T}}).$$

The theorem shows that the cost estimator, the critic, and the actor all converge at a sub-linear rate of $\mathcal{O}(T^{-\frac{1}{2}})$. Correspondingly, to obtain an $\epsilon$-optimal policy, the required sample complexity is $\mathcal{O}(\epsilon^{-2})$. This order is consistent with the existing results on single-timescale AC (Fu et al., 2020; Chen et al., 2021; Olshevsky & Gharesifard, 2022). Nevertheless, our result is the first finite-time analysis of the single-sample single-timescale AC with a global optimality guarantee.

### 4.1 PROOF SKETCH

The main challenge in the finite-time analysis lies in that the estimation errors of the time-average cost, the critic, and the natural policy gradient are strongly coupled. To overcome this issue, we view the propagation of these errors as an interconnected system and analyze them comprehensively. To see the merit of our analysis framework, we sketch the main proof steps of Theorem 4.3 in the following. The supporting propositions and theorems mentioned below can be found in the Appendix.

We define three measures $A(T), B(T), C(T)$ which denote the average values of the cost estimation error, the critic error, and the square norm of the natural policy gradient, respectively:

$$A(T) := \frac{1}{T} \sum_{t=0}^{T-1} \mathbb{E}y_t^2, \ B(T) := \frac{1}{T} \sum_{t=0}^{T-1} \mathbb{E}\|z_t\|^2, \ C(T) := \frac{1}{T} \sum_{t=0}^{T-1} \mathbb{E}\|E_{K_t}\|^2, \tag{22}$$

where $y_t := \eta_t - J(K_t)$ is the cost estimation error and $z_t := \omega_t - \omega_t^*$ with $\omega_t^* := \omega_{K_t}^*$ is the critic error. Note that $E_{K_t} = \nabla_{K_t}^N J(K_t)$ is the natural policy gradient according to (13).

We first derive implicit (coupled) upper bounds for the cost estimation error $y_t$, the critic error $z_t$, and the natural gradient $E_{K_t}$, respectively. After that, we solve an interconnected system of inequalities in terms of $A(T), \ B(T), \ C(T)$ to establish the finite-time convergence.

**Step 1: Cost estimation error analysis.** From the cost estimator update rule (Line 7 of Algorithm 1), we decompose the cost estimation error into:

$$y_{t+1}^2 = (1 - 2\gamma_t)y_t^2 + 2\gamma_t y_t(c_t - J(K_t)) + 2y_t(J(K_t) - J(K_{t+1}))$$
$$+ [J(K_t) - J(K_{t+1}) + \gamma_t(c_t - \eta_t)]^2. \tag{23}$$

The second term on the right hand side of (23) is a noise term introduced by random sampling of the state-action pairs, which reduces to 0 after taking the expectations. The third term is the variation of the moving targets $J(K_t)$ tracked by cost estimator. It is bounded by $y_t, z_t, E_{K_t}$ utilizing

the Lipschitz continuity of $J(K_t)$ (Proposition A.6), the actor update rule (21), and the Cauchy-Schwartz inequality. The last term reflects the variance in cost estimation, which is controlled by a high probability bound of $c_t$ (Proposition A.4).

**Step 2: Critic error analysis**. By the critic update rule (Line 8 of Algorithm 1), we decompose the squared error by (neglecting the projection for the time being)

$$\begin{aligned}
\|z_{t+1}\|^2 =& \|z_t\|^2 + 2\beta_t\langle z_t, \bar{h}(\omega_t, K_t)\rangle + 2\beta_t\Lambda(O_t, \omega_t, K_t) + 2\beta_t\langle z_t, \Delta h(O_t, \eta_t, K_t)\rangle \\
&+ 2\langle z_t, \omega_t^* - \omega_{t+1}^*\rangle + \|\beta_t(h(O_t, \omega_t, K_t) + \Delta h(O_t, \eta_t, K_t)) + (\omega_t^* - \omega_{t+1}^*)\|^2, \quad (24)
\end{aligned}$$

where the definitions of $h, \bar{h}, \Delta h, \Lambda$, and $O_t$ can be found in (27) in the Appendix. The second term on the right hand side of (24) is bounded by $-\mu\|z_t\|^2$, where $\mu$ is a lower bound of $\sigma_{\min}(A_{K_t})$ proved in Proposition A.8. The third term is a random noise introduced by sampling, which reduces to 0 after taking expectation. The fourth term is caused by inaccurate cost and critic estimations, which can be bounded by the norm of $y_t$ and $z_t$. The fifth term tracks the difference between the drifting critic targets. We control it by the Lipschitz continuity of the critic target established in Proposition A.9. The last term reflects the variances of various estimations, which is bounded by the diminishing $\beta_t$.

**Step 3: Natural gradient norm analysis**. From the actor update rule (Line 9 of Algorithm 1) and the almost smoothness property of LQR (Lemma A.11), we derive

$$\begin{aligned}
2\text{Tr}(D_{K_{t+1}}E_{K_t}^\top E_{K_t}) =& \frac{1}{\alpha_t}[J(K_t) - J(K_{t+1})] - 2\text{Tr}(D_{K_{t+1}}(\hat{E}_{K_t} - E_{K_t})^\top E_{K_t}) \\
&+ \alpha_t\text{Tr}(D_{K_{t+1}}\hat{E}_{K_t}^\top(R + B^\top P_{K_t}B)\hat{E}_{K_t}), \quad (25)
\end{aligned}$$

where $\hat{E}_{K_t}$ denotes the estimation of the natural gradient $E_{K_t}$. The first term on the left hand side of (25) can be considered as the scaled square norm of the natural gradient. The first term on the right hand side compares the actor's performances between consecutive updates, which is bounded via Abel summation by parts. The second term evaluates the inaccurate natural gradient estimation, which is then bounded by the critic error $z_t$ and the natural gradient $E_{K_t}$. The last term can be considered as the variance of the perturbed natural gradient update, which is controlled by the diminishing stepsize.

**Step 4: Interconnected iteration system analysis.** Taking the expectation and summing (23), (24), and (25) from 0 to $T - 1$, respectively, we obtain the following interconnected iteration system in terms of $A(T), B(T), C(T)$:

$$\begin{aligned}
A(T) \leq& \mathcal{O}(\frac{1}{\sqrt{T}}) + bB(T) + bC(T), \\
B(T) \leq& \mathcal{O}(\frac{1}{\sqrt{T}}) + d\sqrt{A(T)B(T)} + eC(T), \quad (26) \\
C(T) \leq& \mathcal{O}(\frac{1}{\sqrt{T}}) + g\sqrt{B(T)C(T)},
\end{aligned}$$

where $b, d, e, g$ are positive constants. By solving the above system of inequalities, we further prove that if $bd^2 + bd^2g^2 + 2eg^2 < 1$, then $A(T), B(T), C(T)$ converge at a rate of $\mathcal{O}(T^{-\frac{1}{2}})$. This condition can be easily satisfied by choosing the stepsize ratio $c_\alpha$ to be smaller than a threshold defined in (52).

**Step 5: Global convergence analysis.** To prove the global optimality, we utilize the gradient domination condition of LQR (Lemma A.12),

$$J(K) - J(K^*) \leq \frac{1}{\sigma_{\min}(R)}\|D_{K^*}\|\text{Tr}(E_K^\top E_K).$$

This property shows that the actor performance error can be bounded by the norm of the natural gradient (that is, $\text{Tr}(E_K^\top E_K)$). Since we have proved the average natural gradient norm $C(T)$ converges to zero, summation over both sides of the above inequality yields

$$\min_{0\leq t<T}\mathbb{E}[J(K_t) - J(K^*)] = \mathcal{O}(\frac{1}{\sqrt{T}}),$$

which is the convergence of the actor performance error. We thus complete the proof of Theorem 4.3.

## 5    EXPERIMENTS

We provide two numerical examples to illustrate our theoretical results. The first example is a two-dimensional system and the second example is a four-dimensional system (See Appendix D for the system matrices and other settings). The performance of Algorithm 1 is shown in Figure 1, where the left column corresponds to the first example and the right column to the second example. The solid lines plot the mean values and the shaded regions denote the 95% confidence interval over 10 independent runs. Consistent with our theorem, Figure 1(a) shows that the cost estimation error, the critic error, and the actor performance error all diminish at a rate of at least $T^{-\frac{1}{2}}$. The convergence also suggests that the intermediate closed-loop linear systems during iteration are uniformly stable.

We also compare Algorithm 1 with the zeroth-order method (Fazel et al., 2018) and the double-loop AC algorithm proposed in (Yang et al., 2019) (listed in Algorithm 2 and  Algorithm 3, respectively, in Appendix D). We plotted the relative errors of the actor parameters for all three methods in Figure 1(b). Algorithm 1 demonstrates superior sample-efficiency compared to the other two algorithms, which is well supported by our theoretical analysis.

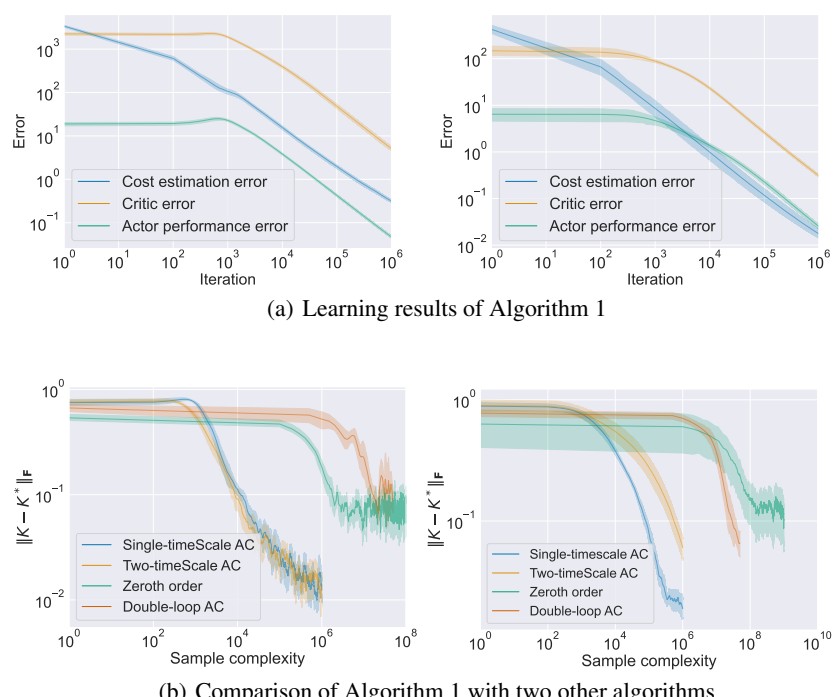

(a) Learning results of Algorithm 1

(b) Comparison of Algorithm 1 with two other algorithms

Figure 1: **(a)** Learning results of Algorithm 1. Here the cost estimation error refers to $\frac{1}{T}\sum_{t=0}^{T-1}(\eta_t - J(K_t))^2$, Critic error refers to $\frac{1}{T}\sum_{t=0}^{T-1}\|\omega_t - \omega_{K_t}^*\|^2$, and the Actor performance error refers to $\frac{1}{T}\sum_{t=0}^{T-1}[J(K_t) - J(K^*)]$, corresponding to the conclusion in Theorem 4.3 empirically. **(b)** Comparison of Algorithm 1 with two other algorithms. The plots are the actor error $\|K - K^*\|_F$.

## 6    CONCLUSION AND DISCUSSION

In this paper, we establish the first finite-time global convergence analysis for the single-sample single-timescale natural actor-critic method under the Linear Quadratic Regulation (LQR) setting. Our work is the first one adopting the simplest single-sample single-timescale structure for solving LQR, which may serve as the first step towards understanding the limits of the AC(NAC) on continuous control task. We provide a novel analysis framework that systematically establishes the convergence of actor and critic simultaneously. Our framework can be extended to analyze other single-timescale reinforcement learning algorithms.

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

## A  PROOF OF MAIN THEOREMS

We choose stepsizes $\alpha_t = \frac{c_\alpha}{\sqrt{1+t}}, \beta_t = \gamma_t = \frac{1}{\sqrt{1+t}}$. Additional constant multipliers $c_\beta, c_\gamma$ can be considered in a similar way. Before proceeding, we define the following notations for the ease of presentation:

$$
\begin{aligned}
\omega_t^* &:= \omega_{K_t}^*, \\
y_t &:= \eta_t - J(K_t), \\
z_t &:= \omega_t - \omega_t^*, \\
O_t &:= (x_t, u_t, x_t', u_t'), \\
\hat{E}_{K_t} &:= \widehat{\nabla_{K_t}^N J(K_t)}, \\
\Delta h(O, \eta, K) &:= [J(K) - \eta]\phi(x, u), \\
h(O, \omega, K) &:= [c(x, u) - J(K) + (\phi(x', u') - \phi(x, u))^\top \omega]\phi(x, u), \\
\bar{h}(\omega, K) &:= \mathbb{E}_{(x,u)\sim(\rho_K, \pi_K)}[[c(x, u) - J(K) + (\phi(x', u') - \phi(x, u))^\top \omega]\phi(x, u)]. \\
\Lambda(O, \omega, K) &:= \langle \omega - \omega_K^*, h(O, \omega, K) - \bar{h}(\omega, K) \rangle.
\end{aligned}
\tag{27}
$$

In the sequel, we establish implicit (coupled) upper bounds for the cost estimator, the critic, and the actor in Theorem A.7, Theorem A.10, and Theorem A.13, respectively. Then we prove the main Theorem 4.3 by solving an interconnected system of inequalities in Appendix A.4.

Before start, we define two notations which are frequently used in our proof.

**Definition A.1.** *For any symmetric matrix $M \in \mathcal{S}^n$, we define the vector $svec(M) \in \mathbb{R}^{\frac{1}{2}(n+1)}$ as*

$$
svec(M) = (m_{11}, \sqrt{2}m_{21}, \cdots, \sqrt{2}m_{n1}, m_{22}, \sqrt{2}m_{32}, \cdots, \sqrt{2}m_{n2}, \cdots, m_{nn})^\top.
$$

*We further define its inverse $smat(\cdot)$ such that*

$$
smat(svec(M)) = M.
$$

### A.1  COST ESTIMATION ERROR ANALYSIS

In this section, we establish an implicit upper bound for the cost estimator $\eta_t$, in terms of the critic error and the natural gradient norm.

We first give an uniform upper bound for the covariance matrix $D_{K_t}$.

**Proposition A.2.** *(Upper bound for covariance matrix). Suppose that Assumption 4.2 holds. The covariance matrix of the stationary distribution $\mathcal{N}(0, D_{K_t})$ induced by the Markov chain in (8) can be upper bounded by*

$$\|D_{K_t}\| \leq \frac{c_1}{1 - (\frac{1+\rho}{2})^2}\|D_\sigma\| \text{ for all } t, \tag{28}$$

*where $c_1$ is a constant.*

Note that the distribution of state-action pair is unbounded so that the feature function is also unbounded. We can establish an upper bound for the tail probability of $(x, u)$ by the Hansen-Wright inequality, the proof of which can be found in Rudelson & Vershynin (2013).

**Lemma A.3.** *(Hansen-Wright inequality). For any integer $m > 0$, let $A$ be a matrix in $\mathbb{R}^{m \times m}$ and let $\eta \sim \mathcal{N}(0, I_m)$ be the standard Gaussian random variable in $\mathbb{R}^m$. Then there exists an absolute constant $\bar{c} > 0$ such that, for any $\theta \geq 0$, we have*

$$\mathbb{P}[|\eta^\top A \eta - \mathbb{E}(\eta^\top A \eta)| > \theta] \leq 2e^{-\bar{c} \cdot min\{\theta^2\|A\|_F^{-2}, \theta\|A\|^{-1}\}}.$$

With this lemma, we can provide an uniform upper bound for the cost under high probability.

**Proposition A.4.** *(Upper bound for cost). With probability at least $1 - 10^{-10}$, for $t = 0, 1, 2, \cdots, T-1$, the cost satisfies*

$$\|x_t\|^2 + \|u_t\|^2 \leq \bar{U},$$
$$c(x_t, u_t) \leq \bar{U},$$

*where*

$$\bar{U} = 2c_2(\sigma_{\max}(Q) + \sigma_{\max}(R) + 1)[\sigma^2 + (1 + \bar{K}^2)\frac{c_1}{1 - (\frac{1+\rho}{2})^2}\|D_\sigma\|]\log(10) \tag{29}$$

*and $c_2$ is a constant.*

Hereafter, we use $\bar{U}$ as an upper bound for all cost $c(x_t, u_t)$. As a consequence, we choose $\eta_0 \leq \bar{U}$ so that we have $\eta_t \leq \bar{U}$ for all $t$.

**Lemma A.5.** *(Perturbation of $P_K$). Suppose $K'$ is a small perturbation of $K$ in the sense that*

$$\|K' - K\| \leq \frac{\sigma_{min}(D_0)}{4}\|D_K\|^{-1}\|B\|^{-1}(\|A - BK\| + 1)^{-1}. \tag{30}$$

*Then we have*

$$\|P_{K'} - P_K\| \leq 6\sigma_{min}^{-1}(D_0)\|D_K\|\|K\|\|R\|(\|K\|\|B\| \cdot \|A - BK\| + \|K\|\|B\| + 1)\|K - K'\|.$$

*Proof.* See Lemma 5.7 in Yang et al. (2019) for a detailed proof. □

With the perturbation of $P_K$, we are ready to prove the Lipschitz continuous of $J(K)$.

**Proposition A.6.** *(Local Lipschitz continuity of $J(K)$) Suppose Lemma A.5 holds, for any $K_t, K_{t+1}$, we have*

$$|J(K_{t+1}) - J(K_t)| \leq l_1\|K_{t+1} - K_t\|,$$

*where*

$$l_1 := 6c_1 d\bar{K}\sigma_{min}^{-1}(D_0)\frac{\|D_\sigma\|^2}{1 - (\frac{1+\rho}{2})^2}\|R\|(\bar{K}\|B\|(\|A\| + \bar{K}\|B\| + 1) + 1). \tag{31}$$

Equipped with the above propositions and lemmas, we are able to bound the cost estimation error.

**Theorem A.7.** *Suppose that Assumptions 4.1 and 4.2 hold and choose $\alpha_t = \frac{c_\alpha}{\sqrt{1+t}}, \beta_t = \gamma_t = \frac{1}{\sqrt{1+t}}$, where $c_\alpha$ is a small positive constant. With probability at least $1 - 10^{-10}$, we have*

$$\frac{1}{T}\sum_{t=0}^{T-1}\mathbb{E}y_t^2 \leq (4l_1^2(\bar{K}+1)^2\bar{\omega}^2c_\alpha^2 + 3\bar{U}^2)\frac{1}{\sqrt{T}} + \frac{l_1c_\alpha}{T}\sum_{t=0}^{T-1}\mathbb{E}\|z_t\|^2 + \frac{l_1c_\alpha}{T}\sum_{t=0}^{T-1}\mathbb{E}\|E_{K_t}\|^2. \tag{32}$$

*Proof.* From line 5 of Algorithm 1, we have

$$
\begin{aligned}
y_{t+1}^2 =&(y_t + J(K_t) - J(K_{t+1}) + \gamma_t(c_t - \eta_t))^2 \\
\leq& y_t^2 + 2\gamma_t y_t(c_t - \eta_t) + 2y_t(J(K_t) - J(K_{t+1})) + 2(J(K_t) - J(K_{t+1}))^2 + 2\gamma_t^2(c_t - \eta_t)^2 \\
=&(1 - 2\gamma_t)y_t^2 + 2\gamma_t y_t(c_t - J(K_t)) + 2\gamma_t^2(c_t - \eta_t)^2 + 2y_t(J(K_t) - J(K_{t+1})) \\
&+ 2(J(K_t) - J(K_{t+1}))^2.
\end{aligned}
$$

Taking expectation up to $(x_t, u_t)$ for both sides, we have

$$
\begin{aligned}
\mathbb{E}[y_{t+1}^2] \leq&(1 - 2\gamma_t)\mathbb{E}y_t^2 + 2\gamma_t\mathbb{E}[y_t(c_t - J(K_t))] + 2\gamma_t^2\mathbb{E}(c_t - \eta_t)^2 + 2\mathbb{E}y_t(J(K_t) - J(K_{t+1})) \\
&+ 2\mathbb{E}(J(K_t) - J(K_{t+1}))^2.
\end{aligned}
$$

To compute $\mathbb{E}[y_t(c_t - J(K_t))]$, we use the notation $v_t$ to denote the vector $(x_t, u_t)$ and $v_{0:t}$ to denote the sequence $(x_0, u_0), (x_1, u_1), \cdots, (x_t, u_t)$. Hence, we have

$$
\mathbb{E}[y_t(c_t - J(K_t))] = \mathbb{E}_{v_{0:t}}[y_t(c_t - J(K_t))] = \mathbb{E}_{v_{0:t-1}}\mathbb{E}_{v_{0:t}}[y_t(c_t - J(K_t))|v_{0:t-1}]
$$

Once we know $v_{0:t-1}$, $y_t$ is not a random variable any more. Thus we get

$$
\begin{aligned}
&\mathbb{E}_{v_{0:t-1}}\mathbb{E}_{v_{0:t}}[y_t(c_t - J(K_t))|v_{0:t-1}] \\
=&\mathbb{E}_{v_{0:t-1}}y_t\mathbb{E}_{v_{0:t}}[(c_t - J(K_t))|v_{0:t-1}] \\
=&\mathbb{E}_{v_{0:t-1}}y_t\mathbb{E}_{v_t}[c_t - J(K_t)|v_{0:t-1}] \\
=&0
\end{aligned}
$$

Hereafter, we need to verify Lemma A.5 first and use the local Lipschitz continuous property of $J(K)$ provided by Proposition A.6 to bound the cost estimation error. Since we have

$$
\|K_{t+1} - K_t\| = \alpha_t\|(\text{smat}(\omega_t)^{22}K_t - \text{smat}(\omega_t)^{21})\|,
$$

to satisfy (30), we choose

$$
c_\alpha \leq \frac{(1 - (\frac{1+\rho}{2})^2)\sigma_{\min}(D_0)}{4c_1\|D_\sigma\|\|B\|(1 + \|A\| + \bar{K}\|B\|)(\bar{K} + 1)\bar{\omega}}. \tag{33}
$$

Hence, according to the update rule, we have

$$
\begin{aligned}
\|K_{t+1} - K_t\| =&\alpha_t\|(\text{smat}(\omega_t)^{22}K_t - \text{smat}(\omega_t)^{21})\| \\
\leq&\frac{c_\alpha}{(1+t)^\delta}(\bar{K}\|\text{smat}(\omega_t)^{22}\| + \|\text{smat}(\omega_t)^{21}\|) \\
\leq&\frac{c_\alpha}{(1+t)^\delta}(\bar{K}\|\omega_t\| + \|\omega_t\|) \\
\leq&\frac{c_\alpha}{(1+t)^\delta}(\bar{K} + 1)\bar{\omega} \\
\leq&\frac{(1 - (\frac{1+\rho}{2})^2)\sigma_{\min}(D_0)}{4c_1\|D_\sigma\|\|B\|(1 + \|A\| + \bar{K}\|B\|)}\frac{1}{(1+t)^\delta} \\
\leq&\frac{\sigma_{\min}(D_0)}{4}\|D_{K_t}\|^{-1}\|B\|^{-1}(\|A - BK_t\| + 1)^{-1}, \tag{34}
\end{aligned}
$$

where the last inequality comes from (28). Thus Lemma A.5 holds for Algorithm 1. As a consequence, Proposition A.6 is also guaranteed.

Combining the fact $2\gamma_t\mathbb{E}[y_t(c_t - J(K_t))] = 0$, we get

$$
\begin{aligned}
\mathbb{E}[y_{t+1}^2] \leq & (1 - 2\gamma_t)\mathbb{E}y_t^2 + 2\mathbb{E}y_t(J(K_t) - J(K_{t+1})) + 2\mathbb{E}(J(K_t) - J(K_{t+1}))^2 + 2\gamma_t^2\mathbb{E}(c_t - \eta_t)^2 \\
\leq & (1 - 2\gamma_t)\mathbb{E}y_t^2 + 2\mathbb{E}|y_t||J(K_t) - J(K_{t+1})| + 2\mathbb{E}(J(K_t) - J(K_{t+1}))^2 + 2\gamma_t^2\mathbb{E}(c_t - \eta_t)^2 \\
\leq & (1 - 2\gamma_t)\mathbb{E}y_t^2 + 2l_1\mathbb{E}|y_t|\|K_t - K_{t+1}\| + 2\mathbb{E}(J(K_t) - J(K_{t+1}))^2 + 2\gamma_t^2\mathbb{E}(c_t - \eta_t)^2 \\
\leq & (1 - 2\gamma_t)\mathbb{E}y_t^2 + 2l_1\alpha_t\mathbb{E}|y_t|\|\widehat{E}_{K_t}\| + 2\mathbb{E}(J(K_t) - J(K_{t+1}))^2 + 2\gamma_t^2\mathbb{E}(c_t - \eta_t)^2 \\
\leq & (1 - 2\gamma_t)\mathbb{E}y_t^2 + 2l_1\alpha_t\mathbb{E}|y_t|\|\widehat{E}_{K_t} - E_{K_t} + E_{K_t}\| + 2\mathbb{E}(J(K_t) - J(K_{t+1}))^2 \\
& + 2\gamma_t^2\mathbb{E}(c_t - \eta_t)^2 \\
\leq & (1 - 2\gamma_t)\mathbb{E}y_t^2 + 2l_1\alpha_t\mathbb{E}[(2\bar{K}^2 + 2)|y_t|\|z_t\| + |y_t|\|E_{K_t}\|] + 2\mathbb{E}(J(K_t) - J(K_{t+1}))^2 \\
& + 2\gamma_t^2\mathbb{E}(c_t - \eta_t)^2 \\
\leq & (1 - 2\gamma_t)\mathbb{E}y_t^2 + 2l_1\alpha_t\mathbb{E}[2(\bar{K}^2 + 1)^2 y_t^2 + \|z_t\|^2/2 + y_t^2/2 + \|E_{K_t}\|^2/2] \\
& + 2\mathbb{E}(J(K_t) - J(K_{t+1}))^2 + 2\gamma_t^2\mathbb{E}(c_t - \eta_t)^2 \\
\leq & (1 - (2\gamma_t - 2l_1\alpha_t(2(\bar{K}^2 + 1)^2 + \frac{1}{2})))\mathbb{E}y_t^2 + l_1\alpha_t\mathbb{E}\|z_t\|^2 + l_1\alpha_t\mathbb{E}\|E_{K_t}\|^2 \\
& + 2\mathbb{E}(J(K_t) - J(K_{t+1}))^2 + 2\gamma_t^2\mathbb{E}(c_t - \eta_t)^2,
\end{aligned}
$$

where we use the fact that

$$
\|\widehat{E}_{K_t} - E_{K_t}\| \leq 2(\bar{K} + 1)\|\omega_t - \omega_t^*\|.
$$

Choose $c_\alpha$ small enough such that

$$
2l_1 c_\alpha(2(\bar{K}^2 + 1)^2 + \frac{1}{2}) \leq 1. \tag{35}
$$

Then we get

$$
\gamma_t \geq 2l_1\alpha_t(2(\bar{K}^2 + 1)^2 + \frac{1}{2}).
$$

Thus we have

$$
\mathbb{E}[y_{t+1}^2] \leq (1 - \gamma_t)\mathbb{E}y_t^2 + l_1\alpha_t\mathbb{E}\|z_t\|^2 + l_1\alpha_t\mathbb{E}\|E_{K_t}\|^2 + 2\mathbb{E}(J(K_t) - J(K_{t+1}))^2 + 2\gamma_t^2\mathbb{E}(c_t - \eta_t)^2
$$

Rearranging and summing from $0$ to $T - 1$, we have

$$
\sum_{t=0}^{T-1}\mathbb{E}y_t^2 \leq \underbrace{\sum_{t=0}^{T-1}\frac{1}{\gamma_t}\mathbb{E}(y_t^2 - y_{t+1}^2)}_{I_1} + \underbrace{\sum_{t=0}^{T-1}\frac{2}{\gamma_t}\mathbb{E}(J(K_t) - J(K_{t+1}))^2}_{I_2} + \underbrace{\sum_{t=0}^{T-1}2\gamma_t\mathbb{E}(c_t - \eta_t)^2}_{I_3}
$$
$$
+ l_1 c_\alpha\sum_{t=0}^{T-1}\mathbb{E}\|z_t\|^2 + l_1 c_\alpha\sum_{t=0}^{T-1}\mathbb{E}\|E_{K_t}\|^2.
$$

In the sequel, we need to control $I_1, I_2, I_3$ respectively. For $I_1$, following Abel summation by parts, we have

$$
\begin{aligned}
I_1 &= \sum_{t=0}^{T-1}\frac{1}{\gamma_t}\mathbb{E}(y_t^2 - y_{t+1}^2) \\
&= \sum_{t=1}^{T-1}(\frac{1}{\gamma_t} - \frac{1}{\gamma_{t-1}})\mathbb{E}(y_t^2) + \frac{1}{\gamma_0}\mathbb{E}(y_0^2) - \frac{1}{\gamma_{T-1}}\mathbb{E}(y_T^2) \\
&\leq \bar{U}^2\sum_{t=1}^{T-1}(\frac{1}{\gamma_t} - \frac{1}{\gamma_{t-1}}) + \frac{1}{\gamma_0}\bar{U}^2 \\
&\leq \frac{\bar{U}^2}{\gamma_{T-1}} \\
&= \bar{U}^2\sqrt{T},
\end{aligned}
$$

where we use the fact that $|y_t| \leq \bar{U}$. For $I_2$, we get

$$I_2 = \sum_{t=0}^{T-1} \frac{2}{\gamma_t} \mathbb{E}(J(K_t) - J(K_{t+1}))^2$$

$$\leq 2l_1^2(\bar{K}+1)^2\bar{\omega}^2 \sum_{t=0}^{T-1} \frac{1}{\gamma_t}\alpha_t^2$$

$$= 2l_1^2(\bar{K}+1)^2\bar{\omega}^2 c_\alpha^2 \sum_{t=0}^{T-1} \frac{1}{\sqrt{(1+t)}}$$

$$\leq 4l_1^2(\bar{K}+1)^2\bar{\omega}^2 c_\alpha^2 \sqrt{T},$$

where the last inequality is due to

$$\sum_{t=0}^{T-1} \frac{1}{\sqrt{(1+t)}} \leq \int_0^T t^{-\frac{1}{2}} \, dt = 2\sqrt{T}.$$

For $I_3$, we have

$$I_3 = \sum_{t=0}^{T-1} \gamma_t \mathbb{E}(c_t - \eta_t)^2$$

$$\leq \sum_{t=0}^{T-1} \gamma_t \bar{U}^2$$

$$\leq 2\bar{U}^2\sqrt{T}.$$

where we use the fact $0 \leq c_t, \eta_t \leq \bar{U}$ derived by Proposition A.4.

Combining all terms together, we get

$$\sum_{t=0}^{T-1} \mathbb{E}y_t^2 \leq (4l_1^2(\bar{K}+1)^2\bar{\omega}^2 c_\alpha^2 + 3\bar{U}^2)\sqrt{T} + l_1 c_\alpha \sum_{t=0}^{T-1} \mathbb{E}\|z_t\|^2 + l_1 c_\alpha \sum_{t=0}^{T-1} \mathbb{E}\|E_{K_t}\|^2.$$

Dividing by $T$, we have

$$\frac{1}{T}\sum_{t=0}^{T-1} \mathbb{E}y_t^2 \leq (4l_1^2(\bar{K}+1)^2\bar{\omega}^2 c_\alpha^2 + 3\bar{U}^2)\frac{1}{\sqrt{T}} + \frac{l_1 c_\alpha}{T} \sum_{t=0}^{T-1} \mathbb{E}\|z_t\|^2 + \frac{l_1 c_\alpha}{T} \sum_{t=0}^{T-1} \mathbb{E}\|E_{K_t}\|^2.$$

Thus we finish our proof. $\qquad\square$

### A.2  CRITIC ERROR ANALYSIS

In this section, we derive an implicit bound for the critic error, in terms of the cost estimator error and the natural gradient norm. First, we need the following propositions.

**Proposition A.8.** *For all the $K_t$, there exists a constant $\mu > 0$ such that*

$$\sigma_{min}(A_{K_t}) \geq \mu.$$

**Proposition A.9.** *(Lipschitz continuity of $\omega_t^*$) For any $\omega_t^*, \omega_{t+1}^*$, we have*

$$\|\omega_t^* - \omega_{t+1}^*\| \leq l_2\|K_t - K_{t+1}\|, \tag{36}$$

*where*

$$l_2 = 6c_1 d^{\frac{3}{2}}\bar{K}(\|A\| + \|B\|)^2\sigma_{min}^{-1}(D_0)\frac{\|D_\sigma\|\|R\|}{1 - (\frac{1+\rho}{2})^2}(\bar{K}\|B\|(\|A\| + \bar{K}\|B\| + 1) + 1). \tag{37}$$

**Theorem A.10.** *Suppose that Assumptions 4.1 and 4.2 hold and choose $\alpha_t = \frac{c_\alpha}{\sqrt{1+t}}, \beta_t = \gamma_t = \frac{1}{\sqrt{1+t}}$, where $c_\alpha$ is a small positive constant. With probability at least $1 - 10^{-10}$, we have*

$$\frac{1}{T} \sum_{t=1}^{T-1} \mathbb{E}\|z_t\|^2 \leq \frac{4}{\mu}(\bar{U}^4(1+2\bar{\omega})^2 + \bar{\omega}^2 + l_2^2 c_3^2)\frac{1}{\sqrt{T}} + \frac{l_2 c_\alpha}{\mu T} \sum_{t=0}^{T-1} \mathbb{E}\|E_{K_t}\|^2$$

$$+ \frac{2}{\mu}\bar{U}(\frac{1}{T}\sum_{t=0}^{T-1}\mathbb{E}y_t^2)^{\frac{1}{2}}(\frac{1}{T}\sum_{t=0}^{T-1}\mathbb{E}\|z_t\|^2)^{\frac{1}{2}}. \tag{38}$$

*Proof.* Since we have $A_{K_t}\omega_t^* = b_{K_t}$, where $b_{K_t} = \mathbb{E}_{(x_t,u_t)}[(c(x_t,u_t) - J(K_t))\phi(x_t,u_t)]$, we can further get

$$\|\omega_t^*\| = \|A_{K_t}^{-1}b_{K_t}\|$$
$$\leq \frac{1}{\mu}\bar{U}\|\phi(x_t,u_t)\|$$
$$\leq \frac{1}{\mu}\bar{U}^2,$$

where in the last inequality, we use the fact that

$$\|\phi(x,u)\| = \|\begin{pmatrix} x \\ u \end{pmatrix}(x^\top \ u^\top)\|_{\mathrm{F}}$$
$$= \|\begin{pmatrix} x \\ u \end{pmatrix}(x^\top \ u^\top)\|$$
$$\leq \mathrm{Tr}(\begin{pmatrix} x \\ u \end{pmatrix}(x^\top \ u^\top))$$
$$= \|x\|^2 + \|u\|^2$$
$$\leq \bar{U}.$$

Hence, we set

$$\bar{\omega} = \frac{1}{\mu}\bar{U}^2 \tag{39}$$

such that all $\omega_t^*$ lie within this projection radius for all $t$.

From update rule of critic in Algorithm 1, we have

$$\omega_{t+1} = \Pi_{\bar{\omega}}(\omega_t + \beta_t\delta_t\phi(x_t,u_t)),$$

which further implies

$$\omega_{t+1} - \omega_{t+1}^* = \Pi_{\bar{\omega}}(\omega_t + \beta_t\delta_t\phi(x_t,u_t)) - \omega_{t+1}^*.$$

By applying 1-Lipschitz continuity of projection map, we have

$$\|\omega_{t+1} - \omega_{t+1}^*\| = \|\Pi_{\bar{\omega}}(\omega_t + \beta_t\delta_t\phi(x_t,u_t)) - \omega_{t+1}^*\|$$
$$= \|\Pi_{\bar{\omega}}(\omega_t + \beta_t\delta_t\phi(x_t,u_t)) - \Pi_{\bar{\omega}}(\omega_{t+1}^*)\|$$
$$\leq \|\omega_t + \beta_t\delta_t\phi(x_t,u_t) - \omega_{t+1}^*\|$$
$$= \|\omega_t - \omega_t^* + \beta_t\delta_t\phi(s_t,a_t) + (\omega_t^* - \omega_{t+1}^*)\|.$$

This means

$$\|z_{t+1}\|^2 \leq \|z_t + \beta_t\delta_t\phi(s_t,a_t) + (\omega_t^* - \omega_{t+1}^*)\|^2$$
$$= \|z_t + \beta_t(h(O_t,\omega_t,K_t) + \Delta h(O_t,\eta_t,K_t)) + (\omega_t^* - \omega_{t+1}^*)\|^2$$
$$= \|z_t\|^2 + 2\beta_t\langle z_t, h(O_t,\omega_t,K_t)\rangle + 2\beta_t\langle z_t, \Delta h(O_t,\eta_t,K_t)\rangle + 2\langle z_t, \omega_t^* - \omega_{t+1}^*\rangle$$
$$+ \|\beta_t(h(O_t,\omega_t,K_t) + \Delta h(O_t,\eta_t,K_t)) + (\omega_t^* - \omega_{t+1}^*)\|^2$$
$$= \|z_t\|^2 + 2\beta_t\langle z_t, \bar{h}(\omega_t,K_t)\rangle + 2\beta_t\Lambda(O_t,\omega_t,K_t) + 2\beta_t\langle z_t, \Delta h(O_t,\eta_t,K_t)\rangle$$
$$+ 2\langle z_t, \omega_t^* - \omega_{t+1}^*\rangle + \|\beta_t(h(O_t,\omega_t,K_t) + \Delta h(O_t,\eta_t,K_t)) + (\omega_t^* - \omega_{t+1}^*)\|^2$$
$$\leq \|z_t\|^2 + 2\beta_t\langle z_t, \bar{h}(\omega_t,K_t)\rangle + 2\beta_t\Lambda(O_t,\omega_t,K_t) + 2\beta_t\langle z_t, \Delta h(O_t,\eta_t,K_t)\rangle$$
$$+ 2\langle z_t, \omega_t^* - \omega_{t+1}^*\rangle + 2\beta_t^2\|h(O_t,\omega_t,K_t) + \Delta h(O_t,\eta_t,K_t))\|^2 + 2\|\omega_t^* - \omega_{t+1}^*\|^2.$$

From Proposition A.8, we know that $\sigma_{\min}(A_{K_t}) \geq \mu$ for all $K_t$. Then we have

$$
\begin{aligned}
\langle z_t, \bar{h}(\omega_t, K_t) \rangle &= \langle z_t, b_{K_t} - A_{K_t}\omega_t \rangle \\
&= \langle z_t, b_{K_t} - A_{K_t}w_t - (b_{K_t} - A_{K_t}\omega_t^*) \rangle \\
&= \langle z_t, -A_{K_t}z_t \rangle \\
&= -z_t^\top A_{K_t} z_t \\
&\leq -\mu\|z_t\|^2,
\end{aligned}
$$

where we use the fact $A_K \omega_{K_t}^* - b_{K_t} = 0$. Hence, we have

$$
\begin{aligned}
\|z_{t+1}\|^2 \leq &(1 - 2\mu\beta_t)\|z_t\|^2 + 2\beta_t\Lambda(O_t, \omega_t, K_t) + 2\beta_t\langle z_t, \Delta h(O_t, \eta_t, K_t)\rangle + 2\langle z_t, \omega_t^* - \omega_{t+1}^*\rangle \\
&+ 2\beta_t^2 \bar{U}^4(1+\bar{\omega})^2 + 2\|\omega_t^* - \omega_{t+1}^*\|^2,
\end{aligned}
$$

where we use the fact that

$$
\begin{aligned}
&\|h(O_t, \omega_t, K_t) + \Delta h(O_t, \eta_t, K_t))\| \\
=&\|(c(x_t, u_t) - \eta_t)\phi(x_t, u_t) + (\phi(x_t', u_t') - \phi(x_t, u_t))^\top \omega_t\phi(x_t, u_t)\| \\
\leq&\|(c(x_t, u_t) - \eta_t)\phi(x_t, u_t)\| + \|(\phi(x_t', u_t') - \phi(x_t, u_t))^\top \omega_t\phi(x_t, u_t)\| \\
\leq&\bar{U}^2 + 2\bar{U}^2\bar{\omega} \\
=&\bar{U}^2(1 + 2\bar{\omega}).
\end{aligned}
$$

Taking expectation up to $(x_t, u_t)$ and noticing that

$$
\begin{aligned}
\mathbb{E}[\Lambda(O_t, \omega_t, K_t)] &= \mathbb{E}_{v_{0:t}}\left[\langle \omega_t - \omega_{K_t}^*, h(O_t, \omega_t, K_t) - \bar{h}(\omega_t, K_t)\rangle\right] \\
&= \mathbb{E}_{v_{0:t-1}} \mathbb{E}_{v_{0:t}}\left[\langle \omega_t - \omega_{K_t}^*, h(O_t, \omega_t, K_t) - \bar{h}(\omega_t, K_t)\rangle | v_{0:t-1}\right] \\
&= \mathbb{E}_{v_{0:t-1}}\langle \omega_t - \omega_{K_t}^*, \mathbb{E}_{v_t}[h(O_t, \omega_t, K_t) - \bar{h}(\omega_t, K_t)|v_{0:t-1}]\rangle \\
&= 0,
\end{aligned}
$$

we get

$$
\begin{aligned}
\mathbb{E}\|z_{t+1}\|^2 \leq &(1 - 2\mu\beta_t)\mathbb{E}\|z_t\|^2 + 2\beta_t\mathbb{E}\langle z_t, \Delta h(O_t, \eta_t, K_t)\rangle + 2\mathbb{E}\langle z_t, \omega_t^* - \omega_{t+1}^*\rangle \\
&+ 2\mathbb{E}\|\omega_t^* - \omega_{t+1}^*\|^2 + 2\bar{U}^4(1+2\bar{\omega})^2\beta_t^2.
\end{aligned} \tag{40}
$$

Therefore, using $\|\Delta h(O_t, \eta_t, K_t)\| \leq \bar{U}|y_t|$, we can further rewrite (40) as

$$
\begin{aligned}
\mathbb{E}\|z_{t+1}\|^2 \leq &(1 - 2\mu\beta_t)\mathbb{E}\|z_t\|^2 + 2\mathbb{E}\langle z_t, \omega_t^* - \omega_{t+1}^*\rangle + 2\bar{U}\beta_t\mathbb{E}|y_t|\|z_t\| \\
&+ 2\beta_t^2\bar{U}^4(1+2\bar{\omega})^2 + 2\mathbb{E}\|\omega_t^* - \omega_{t+1}^*\|^2.
\end{aligned}
$$

Based on (36), we can rewrite the above inequality as

$$
\begin{aligned}
\mathbb{E}\|z_{t+1}\|^2 \leq &(1 - 2\mu\beta_t)\mathbb{E}\|z_t\|^2 + 2\bar{U}\beta_t\mathbb{E}|y_t|\|z_t\| + 2l_2\mathbb{E}\|z_t\|\|K_t - K_{t+1}\| \\
&+ 2\bar{U}^4(1+2\bar{\omega})^2\beta_t^2 + 2l_2^2\mathbb{E}\|K_t - K_{t+1}\|^2 \\
\leq &(1 - 2\mu\beta_t)\mathbb{E}\|z_t\|^2 + 2\bar{U}\beta_t\mathbb{E}|y_t|\|z_t\| + 2l_2\alpha_t\mathbb{E}\|z_t\|\|\widehat{E}_{K_t}\| \\
&+ 2\bar{U}^4(1+2\bar{\omega})^2\beta_t^2 + 2l_2^2\mathbb{E}\|K_t - K_{t+1}\|^2 \\
\leq &(1 - 2\mu\beta_t)\mathbb{E}\|z_t\|^2 + 2\bar{U}\beta_t\mathbb{E}|y_t|\|z_t\| + 2l_2\alpha_t\mathbb{E}\|z_t\|\|\widehat{E}_{K_t} - E_{K_t} + E_{K_t}\| \\
&+ 2\bar{U}^4(1+2\bar{\omega})^2\beta_t^2 + 2l_2^2\mathbb{E}\|K_t - K_{t+1}\|^2 \\
\leq &(1 - 2\mu\beta_t)\mathbb{E}\|z_t\|^2 + 2l_2\alpha_t\mathbb{E}[\|z_t\|\|\widehat{E}_{K_t} - E_{K_t}\| + \|z_t\|\|E_{K_t}\|] \\
&+ 2\bar{U}\beta_t\mathbb{E}|y_t|\|z_t\| + 2\bar{U}^4(1+2\bar{\omega})^2\beta_t^2 + 2l_2^2\mathbb{E}\|K_t - K_{t+1}\|^2 \\
\leq &(1 - 2\mu\beta_t)\mathbb{E}\|z_t\|^2 + 2l_2\alpha_t\mathbb{E}[2(\bar{K}+1)\|z_t\|^2 + \frac{\|z_t\|^2}{2} + \frac{\|E_{K_t}\|^2}{2}] \\
&+ 2\bar{U}\beta_t\mathbb{E}|y_t|\|z_t\| + 2\bar{U}^4(1+2\bar{\omega})^2\beta_t^2 + 2l_2^2\mathbb{E}\|K_t - K_{t+1}\|^2 \\
\leq &(1 - 2\mu\beta_t)\mathbb{E}\|z_t\|^2 + 2\bar{U}\beta_t\mathbb{E}|y_t|\|z_t\| + (4\bar{K}+5)l_2\alpha_t\mathbb{E}\|z_t\|^2 \\
&+ l_2\alpha_t\mathbb{E}\|E_{K_t}\|^2 + 2(\bar{U}^4(1+2\bar{\omega})^2 + l_2^2 c_3^2)\beta_t^2
\end{aligned} \tag{41}
$$

where the second inequality is due to $\|K_t - K_{t+1}\| \leq \frac{c_3}{(1+t)^\delta} = c_3\beta_t$ from (34), where

$$c_3 := \frac{(1 - (\frac{1+\rho}{2})^2)\sigma_{\min}(D_0)}{4c_1\|D_\sigma\|\|B\|(1 + \|A\| + \bar{K}\|B\|)}. \tag{42}$$

Choose $c_\alpha$ small enough such that

$$(4\bar{K} + 5)l_2c_\alpha \leq \mu. \tag{43}$$

Thus we can rewrite 41 as

$$\begin{aligned}
\mathbb{E}\|z_{t+1}\|^2 \leq &(1 - \mu\beta_t)\mathbb{E}\|z_t\|^2 + 2\bar{U}\beta_t\mathbb{E}|y_t|\|z_t\| + l_2\alpha_t\mathbb{E}\|E_{K_t}\|^2 \\
&+ 2(\bar{U}^4(1 + 2\bar{\omega})^2 + l_2^2c_3^2)\beta_t^2
\end{aligned}$$

Rearranging the inequality and summing from 0 to $T - 1$ yields

$$\begin{aligned}
\mu\sum_{t=1}^{T-1}\mathbb{E}\|z_t\|^2 \leq &\sum_{t=0}^{T-1}\frac{1}{\beta_t}\mathbb{E}(\|z_t\|^2 - \|z_{t+1}\|^2) + 2\bar{U}\sum_{t=0}^{T-1}\mathbb{E}|y_t|\|z_t\| + l_2c_\alpha\sum_{t=0}^{T-1}\mathbb{E}\|E_{K_t}\|^2 \\
&+ 2(\bar{U}^4(1 + 2\bar{\omega})^2 + l_2^2c_3^2)\sum_{t=0}^{T-1}\beta_t \\
\leq &\underbrace{\sum_{t=0}^{T-1}\frac{1}{\beta_t}\mathbb{E}(\|z_t\|^2 - \|z_{t+1}\|^2)}_{I_1} + 2\bar{U}\underbrace{\sum_{t=0}^{T-1}\mathbb{E}|y_t|\|z_t\|}_{I_2} + l_2c_\alpha\sum_{t=0}^{T-1}\mathbb{E}\|E_{K_t}\|^2 \\
&+ 4(\bar{U}^4(1 + 2\bar{\omega})^2 + l_2^2c_3^2)\sqrt{T}.
\end{aligned}$$

orc We need to control $I_1$ and $I_2$, respectively.

For term $I_1$, from Abel summation by parts, we have

$$\begin{aligned}
I_1 &= \sum_{t=0}^{T-1}\frac{1}{\beta_t}\mathbb{E}(\|z_t\|^2 - \|z_{t+1}\|^2) \\
&= \sum_{t=1}^{T-1}(\frac{1}{\beta_t} - \frac{1}{\beta_{t-1}})\mathbb{E}\|z_t\|^2 + \frac{1}{\beta_0}\mathbb{E}\|z_0\|^2 - \frac{1}{\beta_{T-1}}\mathbb{E}\|z_T\|^2 \\
&\leq \sum_{t=1}^{T-1}(\frac{1}{\beta_t} - \frac{1}{\beta_{t-1}})\mathbb{E}\|z_t\|^2 + \frac{1}{\beta_0}\mathbb{E}\|z_0\|^2 \\
&\leq 4\bar{\omega}^2(\sum_{t=1}^{T-1}(\frac{1}{\beta_t} - \frac{1}{\beta_{t-1}}) + \frac{1}{\beta_0}) \\
&= 4\bar{\omega}^2\frac{1}{\beta_{T-1}} \\
&= 4\bar{\omega}^2\sqrt{T}.
\end{aligned}$$

For $I_2$, from Cauchy-Schwartz inequality, we have

$$\begin{aligned}
I_2 &= \sum_{t=0}^{T-1}\mathbb{E}|y_t|\|z_t\| \\
&\leq \sum_{t=0}^{T-1}(\mathbb{E}y_t^2)^{\frac{1}{2}}(\mathbb{E}\|z_t\|^2)^{\frac{1}{2}} \\
&\leq (\sum_{t=0}^{T-1}\mathbb{E}y_t^2)^{\frac{1}{2}}(\sum_{t=0}^{T-1}\mathbb{E}\|z_t\|^2)^{\frac{1}{2}}.
\end{aligned}$$

Combining the upper bound of the above two items, we can get

$$\sum_{t=1}^{T-1} \mathbb{E}\|z_t\|^2 \leq \frac{4}{\mu}(\bar{U}^4(1+2\bar{\omega})^2 + \bar{\omega}^2 + l_2^2 c_3^2)\sqrt{T} + \frac{l_2 c_\alpha}{\mu}\sum_{t=0}^{T-1}\mathbb{E}\|E_{K_t}\|^2$$
$$+ \frac{2}{\mu}\bar{U}(\sum_{t=0}^{T-1}\mathbb{E}y_t^2)^{\frac{1}{2}}(\sum_{t=0}^{T-1}\mathbb{E}\|z_t\|^2)^{\frac{1}{2}}.$$

Dividing by $T$, we have

$$\frac{1}{T}\sum_{t=1}^{T-1}\mathbb{E}\|z_t\|^2 \leq \frac{4}{\mu}(\bar{U}^4(1+2\bar{\omega})^2 + \bar{\omega}^2 + l_2^2 c_3^2)\frac{1}{\sqrt{T}} + \frac{l_2 c_\alpha}{\mu T}\sum_{t=0}^{T-1}\mathbb{E}\|E_{K_t}\|^2$$
$$+ \frac{2}{\mu}\bar{U}(\frac{1}{T}\sum_{t=0}^{T-1}\mathbb{E}y_t^2)^{\frac{1}{2}}(\frac{1}{T}\sum_{t=0}^{T-1}\mathbb{E}\|z_t\|^2)^{\frac{1}{2}},$$

which concludes he convergence of critic. $\square$

## A.3 NATURAL GRADIENT NORM ANALYSIS

In this subsection, we derive an implicit bound for the natural gradient norm in terms of the the critic error. Before proceeding, we need the following two lemmas, which characterize two important properties of LQR system.

**Lemma A.11.** *(Almost Smoothness). For any two stable policies $K$ and $K'$, $J(K)$ and $J(K')$ satisfy:*

$$J(K') - J(K) = -2Tr(D_{K'}(K-K')^\top E_K) + Tr(D_{K'}(K-K')^\top(R+B^\top P_K B)(K-K')).$$

**Lemma A.12.** *(Gradient Domination). Let $K^*$ be an optimal policy. Suppose $K$ has finite cost. Then, it holds that*

$$\sigma_{min}(D_0)\|R+B^\top P_K B\|^{-1}Tr(E_K^\top E_K) \leq J(K) - J(K^*) \leq \frac{1}{\sigma_{min}(R)}\|D_{K^*}\|Tr(E_K^\top E_K).$$

**Theorem A.13.** *Suppose that Assumptions 4.1 and 4.2 hold and choose $\alpha_t = \frac{c_\alpha}{\sqrt{1+t}}, \beta_t = \gamma_t = \frac{1}{\sqrt{1+t}}$, where $c_\alpha$ is a small positive constant. With probability at least $1 - 10^{-10}$, we have*

$$\frac{1}{T}\sum_{t=0}^{T-1}\mathbb{E}\|E_{K_t}\|^2 \leq (\frac{\bar{U}+2c_4 c_\alpha^2}{2\sigma_{min}(D_0)c_\alpha})\frac{1}{\sqrt{T}} + \frac{c_5(\bar{K}+1)}{\sigma_{min}(D_0)}(\frac{1}{T}\sum_{t=0}^{T-1}\mathbb{E}\|z_t\|^2)^{\frac{1}{2}}(\frac{1}{T}\sum_{t=0}^{T-1}\mathbb{E}\|E_{K_t}\|)^{\frac{1}{2}}.$$
$$(44)$$

*Proof.* Combining the almost smoothness property, we get

$$J(K_{t+1}) - J(K_t)$$
$$= -2Tr(D_{K_{t+1}}(K_t - K_{t+1})^\top E_{K_t}) + Tr(D_{K_{t+1}}(K_t-K_{t+1})^\top(R+B^\top P_{K_t}B)(K_t-K_{t+1}))$$
$$= -2\alpha_t Tr(D_{K_{t+1}}\hat{E}_{K_t}^\top E_{K_t}) + \alpha_t^2 Tr(D_{K_{t+1}}\hat{E}_{K_t}^\top(R+B^\top P_{K_t}B)\hat{E}_{K_t})$$
$$= -2\alpha_t Tr(D_{K_{t+1}}(\hat{E}_{K_t} - E_{K_t})^\top E_{K_t}) - 2\alpha_t Tr(D_{K_{t+1}}E_{K_t}^\top E_{K_t})$$
$$+ \alpha_t^2 Tr(D_{K_{t+1}}\hat{E}_{K_t}^\top(R+B^\top P_{K_t}B)\hat{E}_{K_t}).$$

By the similar trick to the proof of Proposition A.2, we can bound $P_{K_t}$ by

$$\|P_{K_t}\| \leq \frac{\hat{c}_1}{1-(\frac{1+\rho}{2})^2}\|Q+K^\top RK\|$$
$$\leq \frac{\hat{c}_1(\sigma_{max}(Q)+\bar{K}^2\sigma_{max}(R))}{1-(\frac{1+\rho}{2})^2},$$

where $\hat{c}_1$ is a constant. Hence we further have

$$\text{Tr}(D_{K_{t+1}}\hat{E}_{K_t}^\top(R + B^\top P_{K_t}B)\hat{E}_{K_t})$$

$$\leq d\|D_{K_{t+1}}\|\|R + B^\top P_{K_t}B\|\|\hat{E}_{K_t}\|_\text{F}^2$$

$$\leq d(\bar{K}+1)^2\bar{\omega}^2\frac{c_1\|D_\sigma\|}{1-(\frac{1+\rho}{2})^2}(\sigma_{\max}(R) + \sigma_{\max}^2(B)\frac{\hat{c}_1(\sigma_{\max}(Q) + \bar{K}^2\sigma_{\max}(R))}{1-(\frac{1+\rho}{2})^2}),$$

where we use $\|\hat{E}_{K_t}\|_\text{F} \leq (\bar{K}+1)\bar{\omega}$. Hence we define $c_4$ as follows

$$c_4 := d(\bar{K}+1)^2\bar{\omega}^2\frac{c_1\|D_\sigma\|}{1-(\frac{1+\rho}{2})^2}(\sigma_{\max}(R) + \sigma_{\max}^2(B)\frac{\hat{c}_1(\sigma_{\max}(Q) + \bar{K}^2\sigma_{\max}(R))}{1-(\frac{1+\rho}{2})^2}). \quad (45)$$

Then we get

$$J(K_{t+1}) - J(K_t)$$

$$\leq -2\alpha_t\text{Tr}(D_{K_{t+1}}(\hat{E}_{K_t} - E_{K_t})^\top E_{K_t}) - 2\alpha_t\text{Tr}(D_{K_{t+1}}E_{K_t}^\top E_{K_t}) + c_4\alpha_t^2$$

$$\leq \alpha_t\frac{2c_1d^{\frac{3}{2}}\|D_\sigma\|}{1-(\frac{1+\rho}{2})^2}\|E_{K_t}\|\|\hat{E}_{K_t} - E_{K_t}\| - 2\alpha_t\sigma_{\min}(D_0)\|E_{K_t}\|^2 + c_4\alpha_t^2$$

$$= c_5\alpha_t\|E_{K_t}\|\|\hat{E}_{K_t} - E_{K_t}\| - 2\alpha_t\sigma_{\min}(D_0)\|E_{K_t}\|^2 + c_4\alpha_t^2,$$

where

$$c_5 := \frac{2c_1d^{\frac{3}{2}}\|D_\sigma\|}{1-(\frac{1+\rho}{2})^2}. \quad (46)$$

Taking expectation up to $(x_t, u_t)$ and rearranging the above inequality, we have

$$\mathbb{E}\|E_{K_t}\|^2 \leq \frac{\mathbb{E}[J(K_t) - J(K_{t+1})]}{2\alpha_t\sigma_{\min}(D_0)} + \frac{c_5}{2\sigma_{\min}(D_0)}\mathbb{E}\|E_{K_t}\|\|\hat{E}_{K_t} - E_{K_t}\| + \frac{c_4\alpha_t}{2\sigma_{\min}(D_0)}.$$

Summing over $t$ from 0 to $T - 1$ gives

$$\sum_{t=0}^{T-1}\mathbb{E}\|E_{K_t}\|^2 \leq \underbrace{\sum_{t=0}^{T-1}\frac{\mathbb{E}[J(K_t) - J(K_{t+1})]}{2\alpha_t\sigma_{\min}(D_0)}}_{I_1} + \frac{c_5}{2\sigma_{\min}(D_0))}\underbrace{\sum_{t=0}^{T-1}\mathbb{E}\|E_{K_t}\|\|\hat{E}_{K_t} - E_{K_t}\|}_{I_2}$$

$$+ \frac{c_4c_\alpha}{\sigma_{\min}(D_0)}\sqrt{T}.$$

For term $I_1$, using Abel summation by parts, we have

$$\sum_{t=0}^{T-1}\frac{\mathbb{E}[J(K_t) - J(K_{t+1})]}{2\alpha_t\sigma_{\min}(D_0)}$$

$$= \frac{1}{2\sigma_{\min}(D_0)}(\sum_{t=1}^{T-1}(\frac{1}{\alpha_t} - \frac{1}{\alpha_{t-1}})\mathbb{E}[J(K_t)] + \frac{1}{\alpha_0}\mathbb{E}[J(K_0)] - \frac{1}{\alpha_{T-1}}\mathbb{E}[J(K_T)])$$

$$\leq \frac{\bar{U}}{2\sigma_{\min}(D_0)}(\sum_{t=1}^{T-1}(\frac{1}{\alpha_t} - \frac{1}{\alpha_{t-1}}) + \frac{1}{\alpha_0})$$

$$= \frac{\bar{U}}{2\sigma_{\min}(D_0)}\frac{1}{\alpha_{T-1}}$$

$$= \frac{\bar{U}}{2c_\alpha\sigma_{\min}(D_0)}\sqrt{T}.$$

For term $I_2$, by Cauchy-Schwartz inequality, we have

$$\sum_{t=0}^{T-1}\mathbb{E}\|E_{K_t}\|\|\hat{E}_{K_t} - E_{K_t}\| \leq (\sum_{t=0}^{T-1}\mathbb{E}\|E_{K_t}\|^2)^{\frac{1}{2}}(\sum_{t=0}^{T-1}\mathbb{E}\|\hat{E}_{K_t} - E_{K_t}\|^2)^{\frac{1}{2}}.$$

Combining the results of $I_1$ and $I_2$, we have

$$
\begin{aligned}
\sum_{t=0}^{T-1} \mathbb{E}\|E_{K_t}\|^2 \leq & (\frac{\bar{U} + 2c_4 c_\alpha^2}{2\sigma_{\min}(D_0)c_\alpha})\sqrt{T} + \frac{c_5}{2\sigma_{\min}(D_0)}(\sum_{t=0}^{T-1}\mathbb{E}\|E_{K_t}\|^2)^{\frac{1}{2}}(\sum_{t=0}^{T-1}\mathbb{E}\|\hat{E}_{K_t} - E_{K_t}\|^2)^{\frac{1}{2}} \\
\leq & (\frac{\bar{U} + 2c_4 c_\alpha^2}{2\sigma_{\min}(D_0)c_\alpha})\sqrt{T} + \frac{c_5(\bar{K}+1)}{\sigma_{\min}(D_0)}(\sum_{t=0}^{T-1}\mathbb{E}\|z_t\|^2)^{\frac{1}{2}}(\sum_{t=0}^{T-1}\mathbb{E}\|E_{K_t}\|)^{\frac{1}{2}}.
\end{aligned}
$$

Dividing by $T$, we get

$$
\frac{1}{T}\sum_{t=0}^{T-1}\mathbb{E}\|E_{K_t}\|^2 \leq (\frac{\bar{U} + 2c_4 c_\alpha^2}{2\sigma_{\min}(D_0)c_\alpha})\frac{1}{\sqrt{T}} + \frac{c_5(\bar{K}+1)}{\sigma_{\min}(D_0)}(\frac{1}{T}\sum_{t=0}^{T-1}\mathbb{E}\|z_t\|^2)^{\frac{1}{2}}(\frac{1}{T}\sum_{t=0}^{T-1}\mathbb{E}\|E_{K_t}\|)^{\frac{1}{2}}.
$$

Thus we conclude our proof. $\qquad\square$

### A.4 INTERCONNECTED ITERATION SYSTEM ANALYSIS

From the definition in 22, we have

$$
A(T) = \frac{1}{T}\sum_{t=0}^{T-1}\mathbb{E}y_t^2, B(T) = \frac{1}{T}\sum_{t=0}^{T-1}\mathbb{E}\|z_t\|^2, C(T) = \frac{1}{T}\sum_{t=0}^{T-1}\mathbb{E}\|E_{K_t}\|^2. \tag{47}
$$

In the following, we give an interconnected iteration system analysis with respect to $A(T)$, $B(T)$ and $C(T)$.

**Theorem A.14.** *Combining (32), (38) and (44), we have*

$$
A(T) = \mathcal{O}(\frac{1}{\sqrt{T}}), \ B(T) = \mathcal{O}(\frac{1}{\sqrt{T}}), \ C(T) = \mathcal{O}(\frac{1}{\sqrt{T}}). \tag{48}
$$

*Proof.* From (32), (38) and (44), we have

$$
\begin{aligned}
A(T) \leq & (4l_1^2(\bar{K}+1)^2\bar{\omega}^2 c_\alpha^2 + 3\bar{U}^2)\frac{1}{\sqrt{T}} + l_1 c_\alpha B(T) + l_1 c_\alpha C(T), \\
B(T) \leq & \frac{4}{\mu}(\bar{U}^4(1+2\bar{\omega})^2 + \bar{\omega}^2 + l_2^2 c_3^2)\frac{1}{\sqrt{T}} + \frac{2}{\mu}\bar{U}\sqrt{A(T)B(T)} + \frac{l_2 c_\alpha}{\mu}C(T), \\
C(T) \leq & (\frac{\bar{U} + 2c_4 c_\alpha^2}{2\sigma_{\min}(D_0)c_\alpha})\frac{1}{\sqrt{T}} + \frac{c_5(\bar{K}+1)}{\sigma_{\min}(D_0)}\sqrt{B(T)C(T)}.
\end{aligned}
$$

For simplicity, we denote

$$
\begin{aligned}
a = & (4l_1^2(\bar{K}+1)^2\bar{\omega}^2 c_\alpha^2 + 3\bar{U}^2)\frac{1}{\sqrt{T}}, \\
b = & l_1 c_\alpha, \\
c = & \frac{4}{\mu}(\bar{U}^4(1+2\bar{\omega})^2 + \bar{\omega}^2 + l_2^2 c_3^2)\frac{1}{\sqrt{T}}, \\
d = & \frac{2}{\mu}\bar{U}, \\
e = & \frac{l_2 c_\alpha}{\mu}, \\
f = & (\frac{\bar{U} + 2c_4 c_\alpha^2}{2\sigma_{\min}(D_0)c_\alpha})\frac{1}{\sqrt{T}}, \\
g = & \frac{c_5(\bar{K}+1)}{\sigma_{\min}(D_0)}.
\end{aligned}
$$

Thus we further have

$$
\begin{aligned}
A(T) \leq & a + bB(T) + bC(T), \\
B(T) \leq & c + d\sqrt{A(T)B(T)} + eC(T), \\
C(T) \leq & f + g\sqrt{B(T)C(T)}.
\end{aligned} \tag{49}
$$

Then we have

$$B(T) \leq c + \frac{1}{2}(d^2 A(T) + B(T)) + eC(T),$$
$$B(T) \leq 2c + d^2 A(T) + 2eC(T). \tag{50}$$

For $C(T)$, we get

$$C(T) \leq f + \frac{1}{2}(g^2 B(T) + C(T)),$$
$$C(T) \leq 2f + g^2 B(T) \tag{51}$$

Combining 49, 50 and 51, we have

$$B(T) \leq 2c + d^2(a + bB(T) + b(2f + g^2 B(T))) + 2e(2f + g^2 B(T))$$
$$= 2c + ad^2 + 2bd^2 f + 4ef + (bd^2 + bd^2 g^2 + 2eg^2)B(T)$$

If $bd^2 + bd^2 g^2 + 2eg^2 < 1$, we have

$$B(T) \leq \frac{2c + ad^2 + 2bd^2 f + 4ef}{1 - bd^2 - bd^2 g^2 - 2eg^2}$$

Note that

$$bd^2 + bd^2 g^2 + 2eg^2 = l_1 c_\alpha \frac{4}{\mu^2}\bar{U}^2 + l_1 c_\alpha \frac{4}{\mu^2}\bar{U}^2 \frac{c_5^2(\bar{K}+1)^2}{\sigma_{\min}^2(D_0)} + \frac{2l_2 c_\alpha}{\mu}\frac{c_5^2(\bar{K}+1)^2}{\sigma_{\min}^2(D_0)}$$
$$= c_\alpha(l_1 \frac{4}{\mu^2}\bar{U}^2 + l_1 \frac{4}{\mu^2}\bar{U}^2 \frac{c_5^2(\bar{K}+1)^2}{\sigma_{\min}^2(D_0)} + \frac{2l_2 c_5^2(\bar{K}+1)^2}{\mu\sigma_{\min}^2(D_0)})$$

Thus we can achieve $bd^2 + bd^2 g^2 + 2eg^2 < 1$ by choosing the stepsize ratio smaller than the following threshold:

$$1/(l_1 \frac{4}{\mu^2}\bar{U}^2 + l_1 \frac{4}{\mu^2}\bar{U}^2 \frac{c_5^2(\bar{K}+1)^2}{\sigma_{\min}^2(D_0)} + \frac{2l_2 c_5^2(\bar{K}+1)^2}{\mu\sigma_{\min}^2(D_0)}) \tag{52}$$

Therefore, we get

$$B(T) \leq \frac{2c + ad^2 + 2bd^2 f + 4ef}{1 - bd^2 - bd^2 g^2 - 2eg^2} = \mathcal{O}(\frac{1}{\sqrt{T}}),$$
$$C(T) \leq 2f + g^2 B(T) = \mathcal{O}(\frac{1}{\sqrt{T}}),$$
$$A(T) \leq a + bB(T) + C(T) = \mathcal{O}(\frac{1}{\sqrt{T}}).$$

Thus we have

$$A(T) = \mathcal{O}(\frac{1}{\sqrt{T}}), \ B(T) = \mathcal{O}(\frac{1}{\sqrt{T}}), \ C(T) = \mathcal{O}(\frac{1}{\sqrt{T}}),$$

which concludes the proof. $\qquad\square$

### A.5 GLOBAL CONVERGENCE ANALYSIS

**Proof of Theorem 4.3**

*Proof.* From gradient domination, we know that

$$\mathbb{E}(J(K_t) - J(K^*)) \leq \frac{1}{\sigma_{\min}(R)}\|D_{K^*}\|\mathbb{E}[\text{Tr}(E_{K_t}^\top E_{K_t})] \leq \frac{d\|D_{K^*}\|}{\sigma_{\min}(R)}\mathbb{E}\|E_{K_t}\|^2. \tag{53}$$

From the convergence of $C(T)$, we know that

$$\frac{1}{T}\sum_{t=0}^{T-1} \mathbb{E}\|E_{K_t}\| = \mathcal{O}(\frac{1}{\sqrt{T}})$$

Hence, we have

$$\min_{0 \leq t < T} \frac{d\|D_{K^*}\|}{\sigma_{\min}(R)} \mathbb{E}\|E_{K_t}\|^2 \leq \frac{d\|D_{K^*}\|}{\sigma_{\min}(R)} \frac{1}{T} \sum_{t=0}^{T-1} \mathbb{E}\|E_{K_t}\|^2 = \mathcal{O}(\frac{1}{\sqrt{T}})$$

Therefore, from 53 we get

$$\min_{0 \leq t < T} \mathbb{E}(J(K_t) - J(K^*)) = \mathcal{O}(\frac{1}{\sqrt{T}}).$$

Thus we conclude the proof of Theorem 4.3.

$\square$

## B    PROOF OF PROPOSITIONS

To establish the Proposition 3.1, we need the following lemma, the proof of which can be found in Nagar (1959); Magnus (1978).

**Lemma B.1.** *Let $g \sim \mathcal{N}(0, I_n)$ be the standard Gaussian random variable in $\mathbb{R}^n$ and let $M, N$ be two symmetric matrices. Then we have*

$$\mathbb{E}[g^\top M g g^\top N g] = 2Tr(MN) + Tr(M)Tr(N).$$

**Proof of Proposition 3.1**:

*Proof.* This proposition is a slight modification of lemma 3.2 in Yang et al. (2019) and the proof is inspired by the proof of this lemma.

For any state-action pair $(x, u) \in \mathbb{R}^{d+k}$, we denote the successor state-action pair following policy $\pi_K$ by $(x', u')$. With this notation, as we defined in (7), we have

$$x' = Ax + Bu + \epsilon, \quad u' = -Kx' + \sigma\zeta.$$

where $\epsilon \sim \mathcal{N}(0, D_0)$ and $\zeta \sim \mathcal{N}(0, I_k)$. We further denote $(x, u)$ and $(x', u')$ by $\vartheta$ and $\vartheta'$ respectively. Therefore, we have

$$\vartheta' = L\vartheta + \varepsilon, \tag{54}$$

where

$$L := \begin{bmatrix} A & B \\ -KA & -KB \end{bmatrix} = \begin{bmatrix} I_d \\ -K \end{bmatrix} [A \quad B], \, \varepsilon := \begin{bmatrix} \epsilon \\ -K\epsilon + \sigma\zeta \end{bmatrix}.$$

Therefore, by definition, we have $\varepsilon \sim \mathcal{N}(0, \tilde{D}_0)$ where

$$\tilde{D}_0 = \begin{bmatrix} D_0 & -D_0 K^\top \\ -KD_0 & KD_0 K^\top + \sigma^2 I_k \end{bmatrix}.$$

Since for any two matrices $M$ and $N$, it holds that $\rho(MN) = \rho(NM)$. Then we get $\rho(L) = \rho(A - BK) < 1$. Consequently, the Markov chain defined in (54) have a stationary distribution $\mathcal{N}(0, \tilde{D}_K)$ denoted by $\tilde{\rho}_K$, where $\tilde{D}_K$ is the unique positive definite solution of the following Lyapunov equation

$$\tilde{D}_K = L\tilde{D}_K L^\top + \tilde{D}_0 \tag{55}$$

Meanwhile, from the fact that $x \sim \mathcal{N}(0, D_K)$ and $u = -Kx + \sigma\zeta$, by direct computation we have

$$\tilde{D}_K = \begin{bmatrix} D_K & -D_K K^\top \\ -KD_K & KD_K K^\top + \sigma^2 I_k \end{bmatrix} = \begin{bmatrix} 0 & 0 \\ 0 & \sigma^2 I_k \end{bmatrix} + \begin{bmatrix} I_d \\ -K \end{bmatrix} D_K \begin{bmatrix} I_d \\ -K \end{bmatrix}^\top.$$

From the fact that $\|AB\|_F \leq \|A\|_F\|B\|$ and $\|A\| \leq \|A\|_F$, we have

$$\|\tilde{D}_K\| \leq \|\tilde{D}_K\|_F \leq \sigma^2 k + \|D_K\|(d + \|K\|_F^2).$$

Then we get

$$\mathbb{E}_{(x,u)}[\phi(x,u)\phi(x,u)^\top] = \mathbb{E}_{\vartheta\sim\tilde{\rho}_K}[\phi(\vartheta)\phi(\vartheta)^\top].$$

Let $M, N$ be any two symmetric matrices with appropriate dimension, we have

$$\begin{aligned}
&\text{svec}(M)^\top \mathbb{E}_{\vartheta\sim\tilde{\rho}_K}[\phi(\vartheta)\phi(\vartheta)^\top]\text{svec}(N)\\
&= \mathbb{E}_{\vartheta\sim\tilde{\rho}_K}[\text{svec}(M)^\top \phi(\vartheta)\phi(\vartheta)^\top\text{svec}(N)]\\
&= \mathbb{E}_{\vartheta\sim\tilde{\rho}_K}[\langle\vartheta\vartheta^\top, M\rangle\langle\vartheta\vartheta^\top, N\rangle]\\
&= \mathbb{E}_{\vartheta\sim\tilde{\rho}_K}[\vartheta^\top M\vartheta\vartheta^\top N\vartheta]\\
&= \mathbb{E}_{g\sim\mathcal{N}(0,I_{d+k})}[g^\top \tilde{D}_K^{1/2}M\tilde{D}_K^{1/2}gg^\top \tilde{D}_K^{1/2}N\tilde{D}_K^{1/2}g],
\end{aligned}$$

where $\tilde{D}_K^{1/2}$ is the square root of $\tilde{D}_K$. By applying Lemma B.1, we have

$$\begin{aligned}
&\text{svec}(M)^\top \mathbb{E}_{\vartheta\sim\tilde{\rho}_K}[\phi(\vartheta)\phi(\vartheta)^\top]\text{svec}(N)\\
&= \mathbb{E}_{g\sim\mathcal{N}(0,I_{d+k})}[g^\top \tilde{D}_K^{1/2}M\tilde{D}_K^{1/2}gg^\top \tilde{D}_K^{1/2}N\tilde{D}_K^{1/2}g]\\
&= 2\text{Tr}(\tilde{D}_K^{1/2}M\tilde{D}_K N\tilde{D}_K^{1/2}) + \text{Tr}(\tilde{D}_K^{1/2}M\tilde{D}_K^{1/2})\text{Tr}(\tilde{D}_K^{1/2}N\tilde{D}_K^{1/2})\\
&= 2\langle M, \tilde{D}_K N\tilde{D}_K\rangle + \langle M, \tilde{D}_K\rangle\langle N, \tilde{D}_K\rangle\\
&= \text{svec}(M)^\top (2\tilde{D}_K \otimes_s \tilde{D}_K + \text{svec}(\tilde{D}_K)\text{svec}(\tilde{D}_K)^\top)\text{svec}(N),
\end{aligned}$$

where the last equality follows from the fact that

$$\text{svec}(\frac{1}{2}(NSM^\top + MSN^\top)) = (M \otimes_s N)\text{svec}(S).$$

for any two matrix $M, N$ and a symmetric matrix $S$ (Schacke, 2004). Thus we have

$$\mathbb{E}_{\vartheta\sim\tilde{\rho}_K}[\phi(\vartheta)\phi(\vartheta)^\top] = 2\tilde{D}_K \otimes_s \tilde{D}_K + \text{svec}(\tilde{D}_K)\text{svec}(\tilde{D}_K)^\top. \tag{56}$$

Similarly

$$\begin{aligned}
\phi(\vartheta') &= \text{svec}[(L\vartheta + \varepsilon)(L\vartheta + \varepsilon)^\top]\\
&= \text{svec}(L\vartheta\vartheta^\top L^\top + L\vartheta\varepsilon^\top - \varepsilon\vartheta^\top L^\top + \varepsilon\varepsilon^\top).
\end{aligned}$$

Since $\epsilon$ is independent of $\vartheta$, we get

$$\mathbb{E}_{\vartheta\sim\tilde{\rho}_K}[\phi(\vartheta)\phi(\vartheta')^\top] = \mathbb{E}_{\vartheta\sim\tilde{\rho}_K}[\phi(\vartheta)\text{svec}(L\vartheta\vartheta^\top L^\top + \tilde{D}_0)].$$

By the same argument, we have

$$\begin{aligned}
&\text{svec}(M)^\top \mathbb{E}_{\vartheta\sim\tilde{\rho}_K}[\phi(\vartheta)\phi(\vartheta')^\top]\text{svec}(N)\\
&= \mathbb{E}_{\vartheta\sim\tilde{\rho}_K}[\langle\vartheta\vartheta^\top, M\rangle\langle L\vartheta\vartheta^\top L^\top + \tilde{D}_0, N\rangle]\\
&= \mathbb{E}_{\vartheta\sim\tilde{\rho}_K}[\vartheta^\top M\vartheta\vartheta^\top L^\top NL\vartheta] + \langle M, \tilde{D}_K\rangle\langle\tilde{D}_0, N\rangle]\\
&= \mathbb{E}_{g\in\mathcal{N}(0,I_{d+k})}[g^\top \tilde{D}_K^{\frac{1}{2}}M\tilde{D}_K^{\frac{1}{2}}gg^\top \tilde{D}_K^{\frac{1}{2}}L^\top NL\tilde{D}_K^{\frac{1}{2}}g]\\
&\quad + \langle M, \tilde{D}_K,\rangle\langle\tilde{D}_0, N\rangle]\\
&= 2\text{Tr}(M\tilde{D}_K L^\top NL\tilde{D}_K) + \text{Tr}(M\tilde{D}_K)\text{Tr}(L^\top NL\tilde{D}_K)\\
&\quad + \langle M, \tilde{D}_K\rangle\langle\tilde{D}_0, N\rangle\\
&= 2\langle M, \tilde{D}_K L^\top NL\tilde{D}_K\rangle + \langle M, \tilde{D}_K\rangle\langle L\tilde{D}_K L^\top, N\rangle\\
&\quad + \langle M, \tilde{D}_K\rangle\langle\tilde{D}_0, N\rangle\\
&= 2\langle M, \tilde{D}_K L^\top NL\tilde{D}_K\rangle + \langle M, \tilde{D}_K\rangle\langle\tilde{D}_K, N\rangle\\
&= \text{svec}(M)^\top (2\tilde{D}_K L^\top \otimes_s \tilde{D}_K L^\top\\
&\quad + \text{svec}(\tilde{D}_K)\text{svec}(\tilde{D}_K)^\top)\text{svec}(N),
\end{aligned}$$

where we make use of the Lyapunov equation (55). Thus we get

$$\mathbb{E}_{\vartheta \sim \tilde{\rho}_K}[\phi(\vartheta)\phi(\vartheta')^\top] = 2\tilde{D}_K L^\top \otimes_s \tilde{D}_K L^\top + \text{svec}(\tilde{D}_K)\text{svec}(\tilde{D}_K)^\top. \qquad (57)$$

Therefore, combining (56) and (57), we have

$$A_K = 2(\tilde{D}_K \otimes_s \tilde{D}_K - \tilde{D}_K L^\top \otimes_s \tilde{D}_K L^\top)$$
$$= 2(\tilde{D}_K \otimes_s \tilde{D}_K)(I - L^\top \otimes_s L^\top),$$

where in the last equality we use the fact that

$$(A \otimes_s B)(C \otimes_s D) = \frac{1}{2}(AC \otimes_s BD + AD \otimes_s BC)$$

for any matrices $A, B, C, D$. Since $\rho(L) < 1$, then $I - L^\top \otimes_s L^\top$ is positive definite, which further implies $A_K$ is invertible.

From Bellman equation of $Q_K$, we have

$$\langle \phi(x, u), \text{svec}(\Omega_K) \rangle = c(x, u) - J(K) + \langle \mathbb{E}[\phi(x', u')|x, u], \text{svec}(\Omega_K) \rangle.$$

Multiply each side by $\phi(x, u)$ and take a expectation with respect to $(x, u)$, we get

$$\mathbb{E}[\phi(x, u)(\phi(x, u) - \mathbb{E}[\phi(x', u')|x, u])^\top]\text{svec}(\Omega_K) = \mathbb{E}[\phi(x, u)(c(x, u) - J(K))].$$

We further have

$$\mathbb{E}[\phi(x, u)(\phi(x, u) - \mathbb{E}[\phi(x', u')|x, u])^\top]$$
$$= \mathbb{E}[\phi(x, u)(\phi(x, u) - \phi(x', u'))^\top]$$
$$= A_K,$$

where the first equality comes from the low of total expectation and

$$\mathbb{E}[\phi(x, u)(c(x, u) - J(K))] = b_K$$

Therefore, we get

$$A_K \text{svec}(\Omega_K) = b_K,$$

which implies $\omega_K^* = \text{svec}(\Omega_K)$. Thus we conclude our proof. □

**Proof of Proposition A.2**:

*Proof.* Since $D_{K_t}$ satisfies the Lyapunov equation defined in (10), we have

$$D_{K_t} = \sum_{k=0}^{\infty}(A - BK_t)^k D_\sigma((A - BK_t)^\top)^k.$$

From Assumption 4.2, we know that $\rho(A - BK_t) \le \rho < 1$. Thus for any $\epsilon > 0$, there exists a sub-multiplicative matrix norm $\|\cdot\|_*$ such that

$$\|A - BK_t\|_* \le \rho(A - BK_t) + \epsilon.$$

Choose $\epsilon = \frac{1-\rho}{2}$, we get

$$\|A - BK_t\|_* \le \frac{1+\rho}{2} < 1.$$

Therefore, we can bound the norm of $D_{K_t}$ by

$$\|D_{K_t}\|_* \le \sum_{k=0}^{\infty}\|A - BK_t\|_*^{2k}\|D_\sigma\|_*$$
$$\le \|D_\sigma\|_* \sum_{k=0}^{\infty}(\frac{1+\rho}{2})^{2k}$$
$$\le \|D_\sigma\|_* \frac{1}{1 - (\frac{1+\rho}{2})^2}.$$

Since all norms are equivalent on the finite dimensional Euclidean space, there exists a constant $c_1$ satisfies

$$\|D_{K_t}\| \leq \frac{c_1}{1 - (\frac{1+\rho}{2})^2}\|D_\sigma\|,$$

which concludes our proof. $\qquad\qquad\qquad\qquad\qquad\qquad\qquad\qquad\qquad\qquad\qquad\qquad\qquad\square$

**Proof of Proposition A.4**:

*Proof.* Since $x_t \sim \mathcal{N}(0, D_{K_t})$ and $u_t = -K_t x_t + \sigma\zeta_t$, we denote the joint distribution of $\vartheta_t = (x_t, u_t)$ by $\tilde{\rho}_{K_t} = \mathcal{N}(0, \tilde{D}_{K_t})$ where

$$
\begin{aligned}
\tilde{D}_{K_t} &= \begin{bmatrix} D_{K_t} & -D_{K_t}K_t^\top \\ -K_t D_{K_t} & K_t D_{K_t} K_t^\top + \sigma^2 I_k \end{bmatrix} \\
&= \begin{bmatrix} 0 & 0 \\ 0 & \sigma^2 I_k \end{bmatrix} + \begin{bmatrix} I_d \\ -K_t \end{bmatrix} D_{K_t} \begin{bmatrix} I_d \\ -K_t \end{bmatrix}^\top.
\end{aligned}
\tag{58}
$$

Based on Lemma A.3, for $(x_t, u_t) \sim \mathcal{N}(0, \tilde{D}_{K_t})$ with $\tilde{D}_{K_t}$ defined in (58), we obtain

$$\mathbb{P}[|\|x_t\|_2^2 + \|u_t\|_2^2 - \mathrm{Tr}(\tilde{D}_{K_t})| > \theta] \leq 2e^{-\bar{c}\cdot\min\{\theta^2\|\tilde{D}_{K_t}\|_\mathrm{F}^{-2}, \theta\|\tilde{D}_{K_t}\|^{-1}\}}.$$

Choose $\theta = c_2\log(10)\|\tilde{D}_{K_t}\|$ with $c_2$ sufficiently large such that $\bar{c}c_2 > 12$ and $\theta^2\|\tilde{D}_{K_t}\|_\mathrm{F}^{-2} > \theta\|\tilde{D}_{K_t}\|^{-1}$, where we make use of the fact that $\|\tilde{D}_{K_t}\|_\mathrm{F}$ is bounded. Hence we have the following probability inequality

$$\mathbb{P}[|\|x_t\|_2^2 + \|u_t\|_2^2 - \mathrm{Tr}(\tilde{D}_{K_t})| \leq c_2\log(10)\|\tilde{D}_{K_t}\|] \geq 1 - 2e^{-\bar{c}c_2\log(10)}.$$

We define the following event

$$A_t = \{|\|x_t\|_2^2 + \|u_t\|_2^2 - \mathrm{Tr}(\tilde{D}_{K_t})| \leq c_2\log(10)\|\tilde{D}_{K_t}\|\}.$$

Then we have $\mathbb{P}(A_t) \geq 1 - 2e^{-\bar{c}c_2\log(10)} \geq 1 - 2\cdot 10^{-12} \geq 1 - 10^{-11}$. We further define

$$\bar{A} = \cap_{0\leq t\leq T-1}A_t.$$

Thus we get $\mathbb{P}(\bar{A}) \geq 1 - 10^{-10}$. In the sequel, we only consider the case when $\bar{A}$ holds. That is, for any $0 \leq t \leq T - 1$, we have

$$
\begin{aligned}
\|x_t\|^2 + \|u_t\|^2 &\leq c_2\log(10)\|\tilde{D}_{K_t}\| + \mathrm{tr}(\tilde{D}_{K_t}) \\
&\leq (c_2\log(10) + d + k)\|\tilde{D}_{K_t}\| \\
&\leq 2c_2\log(10)\|\tilde{D}_{K_t}\| \\
&\leq 2c_2\log(10)[\sigma^2 k + (d + \|K_t\|_\mathrm{F}^2)\|D_{K_t}\|],
\end{aligned}
$$

where the third inequality holds since we choose $c_2$ large enough such that $c_2\log(10) \geq d + k$ and the last inequality is due to the fact

$$\|\tilde{D}_{K_t}\| \leq \sigma^2 k + (d + \|K_t\|_\mathrm{F}^2)\|D_{K_t}\|.$$

From Assumption 4.1, we know that $\|K_t\| \leq \bar{K}$, so we have

$$\|x_t\|^2 + \|u_t\|^2 \leq 2c_2\log(10)[\sigma^2 k + d(1 + \bar{K}^2)\|D_{K_t}\|]. \tag{59}$$

From Proposition A.2, we know that

$$\|D_{K_t}\| \leq \frac{c_1}{1 - (\frac{1+\rho}{2})^2}\|D_\sigma\|.$$

Substitute $\|D_{K_t}\|$ into (59), we get

$$\|x_t\|^2 + \|u_t\|^2 \leq 2c_2\log(10)[\sigma^2 k + d(1 + \bar{K}^2)\frac{c_1}{1 - (\frac{1+\rho}{2})^2}\|D_\sigma\|]. \tag{60}$$

Notice that $c(x, u) = x^\top Q x + u^\top R u$, combine with (60), we get

$$c(x_t, u_t) \leq \sigma_{\max}(Q)\|x_t\|^2 + \sigma_{\max}(R)\|u_t\|^2$$

$$\leq 2c_2(\sigma_{\max}(Q) + \sigma_{\max}(R) + 1)[\sigma^2 k + d(1 + \bar{K}^2)\frac{c_1}{1 - (\frac{1+\rho}{2})^2}\|D_\sigma\|]\log(10)$$

$$:= \bar{U}.$$

Thus we can use $\bar{U}$ as an upper bound to both $c(x, u)$ and $\|x_t\|^2 + \|u_t\|^2$, which concludes the proof. $\qquad\square$

**Proof of Proposition A.6**:

*Proof.*

$$|J(K_{t+1}) - J(K_t)|$$
$$= |\mathrm{Tr}((P_{K_{t+1}} - P_{K_t})D_\sigma)|$$
$$\leq d\|D_\sigma\|\|P_{K_{t+1}} - P_{K_t}\|$$
$$\leq 6d\|D_\sigma\|\sigma_{\min}^{-1}(D_0)\|D_{K_t}\|\|K_t\|\|R\|(\|K_t\|\|B\|\|A - BK_t\| + \|K_t\|\|B\| + 1)\|K_{t+1} - K_t\|$$
$$\leq 6c_1 d\bar{K}\sigma_{\min}^{-1}(D_0)\frac{\|D_\sigma\|^2}{1 - (\frac{1+\rho}{2})^2}\|R\|(\bar{K}\|B\|(\|A\| + \bar{K}\|B\| + 1) + 1)\|K_{t+1} - K_t\|$$
$$= l_1\|K_{t+1} - K_t\|,$$

where the second inequality is due to the perturbation of $P_K$ in Lemma A.5 and

$$l_1 := 6c_1 d\bar{K}\sigma_{\min}^{-1}(D_0)\frac{\|D_\sigma\|^2}{1 - (\frac{1+\rho}{2})^2}\|R\|(\bar{K}\|B\|(\|A\| + \bar{K}\|B\| + 1) + 1).$$

Thus we finish our proof. $\qquad\square$

**Proof of Proposition A.8**:

*Proof.* From Proposition 3.1, we know that

$$A_{K_t} = 2(\tilde{D}_{K_t} \otimes_s \tilde{D}_{K_t})(I - L^\top \otimes_s L^\top).$$

By Assumption 4.2, we have $\rho(L) = \rho(A - BK_t) \leq \rho < 1$. Then we have

$$\|A_{K_t}^{-1}\| = \frac{1}{2}\|(I - L^\top \otimes_s L^\top)^{-1}(\tilde{D}_{K_t} \otimes_s \tilde{D}_{K_t})^{-1}\|$$

$$\leq \frac{1}{2}\|(I - L^\top \otimes_s L^\top)^{-1}\|\|(\tilde{D}_{K_t} \otimes_s \tilde{D}_{K_t})^{-1}\|$$

$$\leq \frac{1}{2(1 - \rho^2)}\|\tilde{D}_{K_t}^{-1}\|^2$$

$$= \frac{1}{2(1 - \rho^2)\sigma_{\min}^2(\tilde{D}_{K_t})}.$$

To bound $\sigma_{\min}(\tilde{D}_{K_t})$, for any $a \in \mathbb{R}^d$ and $b \in \mathbb{R}^k$, we have

$$\begin{pmatrix} a^\top & b^\top \end{pmatrix} \tilde{D}_{K_t} \begin{pmatrix} a \\ b \end{pmatrix}$$

$$= \mathbb{E}_{(x,u)\sim\mathcal{N}(0,\tilde{D}_{K_t})}[\begin{pmatrix} a^\top & b^\top \end{pmatrix} \begin{pmatrix} x \\ u \end{pmatrix} \begin{pmatrix} x^\top & u^\top \end{pmatrix} \begin{pmatrix} a \\ b \end{pmatrix}]$$

$$= \mathbb{E}_{(x,u)\sim\mathcal{N}(0,\tilde{D}_{K_t})}[((a^\top - b^\top K_t)x + \sigma b^\top \zeta) \cdot ((a^\top - b^\top K_t)x + \sigma b^\top \zeta)^\top]$$

$$= \mathbb{E}_{x\sim\mathcal{N}(0,D_{K_t}),\zeta\sim\mathcal{N}(0,I_k)}[(a^\top - b^\top K_t)xx^\top(a - K_t^\top b) + \sigma^2 b^\top \zeta\zeta^\top b]$$

$$\geq \sigma_{\min}(D_{K_t})\|a - K_t^\top b\|^2 + \sigma^2\|b\|^2.$$

For $\|a - K_t^\top b\|^2$, we have

$$
\begin{aligned}
\|a - K_t^\top b\|^2 &\geq \|a\|^2 + \|K_t^\top b\|^2 - 2\|a\|\|K_t^\top\|\|b\| \\
&\geq \|a\|^2 - 2\bar{K}\|a\|\|b\| \\
&\geq \|a\|^2 - \frac{1}{2}(\|a\|^2 + 4\bar{K}^2\|b\|^2) \\
&= \frac{1}{2}\|a\|^2 - 2\bar{K}^2\|b\|^2.
\end{aligned}
$$

Hence we get

$$
\begin{aligned}
&\begin{pmatrix} a^\top & b^\top \end{pmatrix} \tilde{D}_{K_t} \begin{pmatrix} a \\ b \end{pmatrix} \\
&\geq \sigma_{\min}(D_{K_t})\|a - K_t^\top b\|^2 + \sigma^2\|b\|^2 \\
&\geq \sigma_{\min}(D_{K_t})(\frac{1}{2}\|a\|^2 - 2\bar{K}^2\|b\|^2) + \sigma^2\|b\|^2 \\
&\geq \min\{\sigma_{\min}(D_0), \frac{\sigma^2}{4\bar{K}^2}\}(\frac{1}{2}\|a\|^2 - 2\bar{K}^2\|b\|^2) + \sigma^2\|b\|^2 \\
&\geq \min\{\frac{\sigma_{\min}(D_0)}{2}, \frac{\sigma^2}{8\bar{K}^2}, \frac{\sigma^2}{2}\}(\|a\|^2 + \|b\|^2).
\end{aligned}
$$

Thus we have

$$
\sigma_{\min}(\tilde{D}_{K_t}) \geq \min\{\frac{\sigma_{\min}(D_0)}{2}, \frac{\sigma^2}{8\bar{K}^2}, \frac{\sigma^2}{2}\} > 0,
$$

which further implies

$$
\begin{aligned}
\|A_{K_t}^{-1}\| &\leq \frac{1}{2(1-\rho^2)\sigma_{\min}^2(\tilde{D}_{K_t})} \\
&\leq \frac{1}{2(1-\rho^2)(\min\{\frac{\sigma_{\min}(D_0)}{2}, \frac{\sigma^2}{8\bar{K}^2}, \frac{\sigma^2}{2}\})^2}.
\end{aligned}
$$

We define

$$
\mu := 2(1-\rho^2)(\min\{\frac{\sigma_{\min}(D_0)}{2}, \frac{\sigma^2}{8\bar{K}^2}, \frac{\sigma^2}{2}\})^2
$$

such that we get

$$
\sigma_{\min}(A_{K_t}) \geq \mu,
$$

which concludes the proof. $\qquad\square$

**Proof of Proposition A.9:**

*Proof.*

$$\|\omega_t^* - \omega_{t+1}^*\|$$
$$=\|\mathrm{svec}(\Omega_{K_t} - \Omega_{K_{t+1}})\|$$
$$=\|\Omega_{K_t} - \Omega_{K_{t+1}}\|_{\mathrm{F}}$$
$$=\left\| \begin{bmatrix} A^\top(P_{K_t} - P_{K_{t+1}})A & A^\top(P_{K_t} - P_{K_{t+1}})B \\ B^\top(P_{K_t} - P_{K_{t+1}})A & B^\top(P_{K_t} - P_{K_{t+1}})B \end{bmatrix} \right\|_{\mathrm{F}}$$
$$=\|A^\top(P_{K_t} - P_{K_{t+1}})A\|_{\mathrm{F}} + \|A^\top(P_{K_t} - P_{K_{t+1}})B\|_{\mathrm{F}}$$
$$\quad + \|B^\top(P_{K_t} - P_{K_{t+1}})A\|_{\mathrm{F}} + \|B^\top(P_{K_t} - P_{K_{t+1}})B\|_{\mathrm{F}}$$
$$\leq d^{\frac{3}{2}}(\|A\| + \|B\|)^2\|P_{K_t} - P_{K_{t+1}}\|$$
$$\leq 6d^{\frac{3}{2}}(\|A\| + \|B\|)^2\sigma_{\min}^{-1}(D_0)\|D_{K_t}\|\|K_t\|\|R\|\cdot$$
$$\quad (\|K_t\|\|B\|\|A - BK_t\| + \|K_t\|\|B\| + 1)\|K_{t+1} - K_t\|$$
$$\leq 6c_1 d^{\frac{3}{2}}(\|A\| + \|B\|)^2\sigma_{\min}^{-1}(D_0)\frac{\|D_\sigma\|}{1 - (\frac{1+\rho}{2})^2}\bar{K}\|R\|\cdot$$
$$\quad (\bar{K}\|B\|(\|A\| + \bar{K}\|B\| + 1) + 1)\|K_{t+1} - K_t\|$$
$$=l_2\|K_{t+1} - K_t\|,$$

where

$$l_2 := 6c_1 d^{\frac{3}{2}}\bar{K}(\|A\| + \|B\|)^2\sigma_{\min}^{-1}(D_0)\frac{\|D_\sigma\|\|R\|}{1 - (\frac{1+\rho}{2})^2} \cdot (\bar{K}\|B\|(\|A\| + \bar{K}\|B\| + 1) + 1). \quad (61)$$

$\square$

## C  PROOF OF AUXILIARY LEMMAS

The following lemmas are well known and have been established in several papers (Yang et al., 2019; Fazel et al., 2018). We include the proof here only for completeness.

**Proof of Lemma 2.1**:

*Proof.* Since we focus on the family of linear-Gaussian policies defined in (7), we have

$$J(K) = \mathbb{E}_{(x,u)}[c(x,u)]$$
$$= \mathbb{E}_{(x,u)}[x^\top Qx + u^\top Ru]$$
$$= \mathbb{E}_{(x,u)}[x^\top Qx + (-Kx + \sigma\zeta)^\top R(-Kx + \sigma\zeta)]$$
$$= \mathbb{E}_{x\sim\rho_K}\mathbb{E}_{\zeta\sim I_k}[x^\top(Q + K^\top RK)x - \sigma x^\top K^\top R\zeta$$
$$\quad -\sigma\zeta^\top RKx + \sigma^2\zeta^\top R\zeta]$$
$$= \mathbb{E}_{x\sim\rho_K}[x^\top(Q + K^\top RK)x] + \sigma^2\mathrm{Tr}(R)$$
$$= \mathrm{Tr}((Q + K^\top RK)D_K) + \sigma^2\mathrm{Tr}(R). \quad (62)$$

Furthermore, for $K \in \mathbb{R}^{k \times d}$ such that $\rho(AB - K) < 1$ and positive definite matrix $S \in \mathbb{R}^{d \times d}$, we define the following two operators

$$\Gamma_K(S) = \sum_{t\geq 0}(A - BK)^t S[(A - BK)^t]^\top,$$
$$\Gamma_K^\top(S) = \sum_{t\geq 0}[(A - BK)^t]^\top S(A - BK)^t. \quad (63)$$

Hence, $\Gamma_K(S)$ and $\Gamma_K^\top(S)$ satisfy Lyapunov equations

$$\Gamma_K(S) = S + (A - BK)\Gamma_K(S)(A - BK)^\top, \quad (64)$$
$$\Gamma_K^\top(S) = S + (A - BK)^\top\Gamma_K^\top(S)(A - BK) \quad (65)$$

respectively. Therefore, for any positive definite matrices $S_1$ and $S_2$, we get

$$
\begin{aligned}
\mathrm{Tr}(S_1 \Gamma_K(S_2)) &= \sum_{t \geq 0} \mathrm{Tr}(S_1 (A - BK)^t S_2 [(A - BK)^t]^\top) \\
&= \sum_{t \geq 0} \mathrm{Tr}([(A - BK)^t]^\top S_1 (A - BK)^t S_2) \\
&= \mathrm{Tr}(\Gamma_K^\top(S_1) S_2).
\end{aligned}
$$

Combining (10), (55), (64) and (65), we know that

$$
D_K = \Gamma_K(D_\sigma), \quad P_K = \Gamma_K^\top(Q + K^\top R K). \tag{66}
$$

Thus (62) implies

$$
\begin{aligned}
J(K) &= \mathrm{Tr}((Q + K^\top R K) D_K) + \sigma^2 \mathrm{Tr}(R) \\
&= \mathrm{Tr}((Q + K^\top R K) \Gamma_K(D_\sigma)) + \sigma^2 \mathrm{Tr}(R) \\
&= \mathrm{Tr}(\Gamma_K^\top(Q + K^\top R K) D_\sigma) + \sigma^2 \mathrm{Tr}(R) \\
&= \mathrm{Tr}(P_K D_\sigma) + \sigma^2 \mathrm{Tr}(R).
\end{aligned}
$$

It remains to establish the gradient of $J(K)$. Based on (62), we have

$$
\nabla_K J(K) = \nabla_K \mathrm{Tr}((Q + K^\top R K) C))|_{C = D_K} + \nabla_K \mathrm{Tr}(C D_K)|_{C = Q + K^\top R K},
$$

where we use $C$ to denote that we compute the gradient with respect to $K$ and then substitute the expression of $C$. Hence we get

$$
\nabla_K J(K) = 2 R K D_K + \nabla_K \mathrm{Tr}(C_0 D_K)|_{C_0 = Q + K^\top R K}. \tag{67}
$$

Furthermore, we have

$$
\begin{aligned}
&\nabla_K \mathrm{Tr}(C_0 D_K) \\
=&\nabla_K \mathrm{Tr}(C_0 \Gamma_K(D_\sigma)) \\
=&\nabla_K \mathrm{Tr}(C_0 D_\sigma + C_0 (A - BK) \Gamma_K(D_\sigma)(A - BK)^\top) \\
=&\nabla_K \mathrm{Tr}(C_0 D_\sigma) + \nabla_K \mathrm{Tr}((A - BK)^\top C_0 (A - BK) \Gamma_K(D_\sigma)) \\
=& -2 B^\top C_0 (A - BK) \Gamma_K(D_\sigma) + \nabla_K \mathrm{Tr}(C_1 \Gamma_K(D_\sigma))|_{C_1 = (A - BK)^\top C_0 (A - BK)}.
\end{aligned}
$$

Then it reduces to compute $\nabla_K \mathrm{Tr}(C_1 \Gamma_K(D_\sigma))|_{C_1 = (A - BK)^\top C_0 (A - BK)}$. Applying this iteration for $n$ times, we get

$$
\begin{aligned}
&\nabla_K \mathrm{Tr}(C_0 D_K) \\
=& -2 B^\top \sum_{t=0}^n C_t (A - BK) \Gamma_K(D_\sigma) \\
&+ \nabla_K \mathrm{Tr}(C_n \Gamma_K(D_\sigma))|_{C_n = [(A - BK)^n]^\top C_0 (A - BK)^n}.
\end{aligned} \tag{68}
$$

Meanwhile, by Lyapunov equation defined in (11), we have

$$
\begin{aligned}
\sum_{t=0}^\infty C_t &= \sum_{t=0}^\infty [(A - BK)^t]^\top (Q + K^\top R K)(A - BK)^t \\
&= P_K.
\end{aligned}
$$

Since $\rho(A - BK) < 1$, we further get

$$
\begin{aligned}
&\lim_{n \to \infty} \mathrm{Tr}(C_n \Gamma_K(D_\sigma)) \\
\leq& \lim_{n \to \infty} \|(Q + K^\top R K)\| \rho(A - BK)^{2n} \mathrm{Tr}(\Gamma_K(D_\sigma)) \\
=& 0.
\end{aligned}
$$

Thus by letting $n$ go to infinity in (68), we get

$$\nabla_K \text{Tr}(C_0 D_K)|_{C_0 = Q + K^\top R K}$$
$$= -2B^\top P_K (A - BK) \Gamma_K (D_\sigma)$$
$$= -2B^\top P_K (A - BK) D_K.$$

Hence, combining (67), we have

$$\nabla_K J(K) = 2RK D_K - 2B^\top P_K (A - BK) D_K$$
$$= 2[(R + B^\top P_K B)K - B^\top P_K A] D_K,$$

which concludes our proof. $\qquad\square$

**Proof of Lemma 2.2**:

*Proof.* By definition, we have the state-value function as follows

$$V_\theta(x) := \sum_{t=0}^\infty \mathbb{E}_\theta[(c(x_t, u_t) - J(\theta))|x_0 = x]$$
$$= \mathbb{E}_{u \sim \pi_\theta(\cdot|x)}[Q_\theta(x, u)], \qquad (69)$$

Therefore, we have

$$V_K(x) = \sum_{t=0}^\infty \mathbb{E}[c(x_t, u_t) - J(K)|x_0 = x, u_t = -Kx_t + \sigma\zeta_t]$$
$$= \sum_{t=0}^\infty \mathbb{E}\{[x_t^\top (Q + K^\top RK)x_t] + \sigma^2 \text{Tr}(R) - J(K)\}. \qquad (70)$$

Combining the linear dynamic system in (8) and the form of (70), we see that $V_K(x)$ is a quadratic function, which can be denoted by

$$V_K(x) = x^\top P_K x + C_K,$$

where $P_K$ is defined in (11) and $C_K$ only depends on $K$. Moreover, by definition, we know that $\mathbb{E}_{x \sim \rho_K}[V_K(x)] = 0$, which implies

$$\mathbb{E}_{x \sim \rho_K}[x^\top P_K x + C_K] = \text{Tr}(P_K D_K) + C_K = 0.$$

Thus we have $C_K = -\text{Tr}(P_K D_K)$. Hence, the expression of $V_K(x)$ is given by

$$V_K(x) = x^\top P_K x - \text{Tr}(P_K D_K).$$

Therefore, the action-value function $Q_K(x, u)$ can be written as

$$Q(x, u) = c(x, u) - J(K) + \mathbb{E}[V_K(x')|x, u]$$
$$= c(x, u) - J(K) + (Ax + Bu)^\top P_K (Ax + Bu) + \text{Tr}(P_K D_0) - \text{Tr}(P_K D_K)$$
$$= x^\top Q x + u^\top R u + (Ax + Bu)^\top P_K (Ax + Bu) - \sigma^2 \text{Tr}(R + P_K BB^\top) - \text{Tr}(P_K \Sigma_K).$$

Thus we finish the proof. $\qquad\square$

**Proof of Lemma A.11**:

*Proof.* By the definition of operator in (63) and (66), we have

$$x^\top P_{K'} x$$
$$= x^\top \Gamma_{K'}^\top (Q + K'^\top RK')x$$
$$= \sum_{t \geq 0} x^\top [(A - BK')^t]^\top (Q + K'^\top RK')(A - BK')^t x.$$

Hereafter, we define $(A - BK')^t x = x'_t$ and $u'_t = -K'x'_t$. Hence, we further have

$$x^\top P_{K'} x = \sum_{t \geq 0} x'^\top_t (Q + K'^\top RK')x'_t$$
$$= \sum_{t \geq 0} (x'^\top_t Qx'_t + u'^\top_t Ru'_t).$$

Therefore, we get

$$x^\top P_{K'} x - x^\top P_K x$$
$$= \sum_{t \geq 0} [(x'^\top_t Qx'_t + u'^\top Ru'_t) + x'^\top_t P_K x'_t - x'^\top_t P_K x'_t] - x'^\top_0 P_K x'_0$$
$$= \sum_{t \geq 0} [(x'^\top_t Qx'_t + u'^\top Ru'_t) + x'^\top_{t+1} P_K x'_{t+1} - x'^\top_t P_K x'_t]$$
$$= \sum_{t \geq 0} [(x'^\top_t Qx'_t + u'^\top_t Ru'_t) + [(A - BK')x'_t]^\top P_K (A - BK')x'_t - x'_t P_K x'_t]$$
$$= \sum_{t \geq 0} \{x'^\top_t [Q + (K' - K + K)^\top R(K' - K + K)]x'_t$$
$$+ x'^\top_t [A - BK - B(K' - K)]^\top P_K [A - BK - B(K' - K)]x'_t - x'_t P_K x'_t\}$$
$$= \sum_{t \geq 0} \{2x^\top_t (K' - K)^\top [(R + B^\top P_K B)K - B^\top P_K A]x'_t$$
$$+ x'^\top_t (K' - K)^\top (R + B^\top P_K B)(K' - K)x'_t\}$$
$$= \sum_{t \geq 0} [2x'^\top_t (K' - K)^\top E_K x'_t + x'^\top_t (K' - K)^\top (R + B^\top P_K B)(K' - K)x'_t].$$

Define

$$A_{K,K'}(x) := 2x^\top (K' - K)^\top E_K x + x^\top (K' - K)^\top (R + B^\top P_K B)(K' - K)x. \quad (71)$$

Then, from the expression of $J(K)$ in (12a), we have

$$J(K') - J(K)$$
$$= \mathbb{E}_{x \sim \mathcal{N}(0, D_\sigma)}[x^\top (P_{K'} - P_K)x]$$
$$= \mathbb{E}_{x'_0 \sim \mathcal{N}(0, D_\sigma)} \sum_{t \geq 0} A_{K,K'}(x_t)$$
$$= \mathbb{E}_{x'_0 \sim \mathcal{N}(0, D_\sigma)} \sum_{t \geq 0} [2x'^\top_t (K' - K)^\top E_K x'_t + x'^\top_t (K' - K)^\top (R + B^\top P_K B)(K' - K)x'_t]$$
$$= \text{Tr}(2\mathbb{E}_{x'_0 \sim \mathcal{N}(0, D_\sigma)}[\sum_{t \geq 0} x'^\top_t x'_t](K' - K)^\top E_K) +$$
$$\text{Tr}(\mathbb{E}_{x'_0 \sim \mathcal{N}(0, D_\sigma)}[\sum_{t \geq 0} x'^\top_t x'_t](K' - K)^\top (R + B^\top P_K B)(K' - K))$$
$$= -2\text{Tr}(D_{K'}(K - K')^\top E_K) + \text{Tr}(D_{K'}(K - K')^\top (R + B^\top P_K B)(K - K')).$$

where the last equation is due to the fact that

$$\mathbb{E}_{x'_0 \sim \mathcal{N}(0, D_\sigma)}[\sum_{t \geq 0} x'_t (x'_t)^\top]$$
$$= \mathbb{E}_{x \sim \mathcal{N}(0, D_\sigma)} \{\sum_{t \geq 0} (A - BK')^t xx^\top [(A - BK')^t]^\top\}$$
$$= \Gamma_{K'}(D_\sigma) = D_{K'}.$$

Hence, we finish our proof. □

**Proof of Lemma A.12**:

*Proof.* By definition of $A_{K,K'}$ in (71), we have

$$
\begin{aligned}
&A_{K,K'}(x)\\
=&2x^\top(K'-K)^\top E_K x + x^\top (K'-K)^\top(R+B^\top P_K B)(K'-K)x\\
=&\text{Tr}(xx^\top[K'-K+(R+B^\top P_K B)^{-1}E_K]^\top \cdot\\
&(R+B^\top P_K B)[K'-K+(R+B^\top P_K B)^{-1}E_K])\\
&-\text{Tr}(xx^\top E_K^\top(R+B^\top P_K B)^{-1}E_K)\\
\geq&-\text{Tr}(xx^\top E_K^\top(R+B^\top P_K B)^{-1}E_K),
\end{aligned}
$$

where the equality is satisfied when $K'=K-(R+B^\top P_K B)^{-1}E_K$. Therefore, we have

$$
\begin{aligned}
J(K)-J(K^*)&=-\mathbb{E}_{x_0'\sim\mathcal{N}(0,D_\sigma)}\sum_{t\geq 0}A_{K,K^*}(x_t)\\
&\leq\text{Tr}(D_{K^*}E_K^\top(R+B^\top P_K B)^{-1}E_K)\\
&\leq\|D_{K^*}\|\text{Tr}(E_K^\top(R+B^\top P_K B)^{-1}E_K)\\
&\leq\|D_{K^*}\|\|(R+B^\top P_K B)^{-1}\|\text{Tr}(E_K^\top E_K)\\
&\leq\frac{1}{\sigma_{\min}(R)}\|D_{K^*}\|\text{Tr}(E_K^\top E_K).
\end{aligned}
$$

Thus we complete the proof of upper bound.

It remains to establish the lower bound. Since the equality is attained at $K'=K-(R+B^\top P_K B)^{-1}E_K$, we choose this $K'$ such that

$$
\begin{aligned}
J(K)-J(K^*)&\geq J(K)-J(K')\\
&=-\mathbb{E}_{x_0'\sim\mathcal{N}(0,D_\sigma)}[\sum_{t\geq 0}A_{K,K'}(x_t')]\\
&=\text{Tr}(D_{K'}E_K^\top(R+B^\top P_K B)^{-1}E_K)\\
&\geq\sigma_{\min}(D_0)\|R+B^\top P_K B\|^{-1}\text{Tr}(E_K^\top E_K).
\end{aligned}
$$

Overall, we have

$$
\begin{aligned}
&\sigma_{\min}(D_0)\|R+B^\top P_K B\|^{-1}\text{Tr}(E_K^\top E_K)\leq J(K)-J(K^*)\\
&\leq\frac{1}{\sigma_{\min}(R)}\|D_{K^*}\|\text{Tr}(E_K^\top E_K),
\end{aligned}
$$

which concludes our proof. $\qquad\square$

## D  EXPERIMENTAL DETAILS

**Example D.1.** *Consider a two-dimensional system with*

$$
A=\begin{bmatrix}0 & 1\\ 1 & 0\end{bmatrix}, B=\begin{bmatrix}0 & 1\\ 1 & 0\end{bmatrix}, Q=\begin{bmatrix}9 & 2\\ 2 & 1\end{bmatrix}, R=\begin{bmatrix}1 & 2\\ 2 & 8\end{bmatrix}.
$$

**Example D.2.** *Consider a four-dimensional system with*

$$
A=\begin{bmatrix}0.2 & 0.1 & 1 & 0\\ 0.2 & 0.1 & 0.1 & 0\\ 0 & 0.1 & 0.5 & 0\\ 0 & 0 & 0 & 0.5\end{bmatrix}, B=\begin{bmatrix}0.3 & 0 & 0\\ 0.2 & 0 & 0.3\\ 1 & 1 & 0.3\\ 0.3 & 0.1 & 0.1\end{bmatrix},
$$

$$
Q=\begin{bmatrix}1 & 0 & 0.2 & 0\\ 0 & 1 & 0.1 & 0\\ 0.2 & 0.1 & 1 & 0.1\\ 0 & 0 & 0.1 & 1\end{bmatrix}, R=\begin{bmatrix}1 & 0.1 & 1\\ 0.1 & 1 & 0.5\\ 1 & 0.5 & 2\end{bmatrix}.
$$

We compare our considered single-sample single-timescale AC with two other baseline algorithms that have been analyzed in the state-of-the-art theoretical works: the zeroth-order method (Fazel et al., 2018) (listed in Algorithm 2 on the next page) and the double loop AC (Yang et al., 2019) (listed in Algorithm 3 on the next page).

For the considered single-sample single-timescale AC, we set for both examples $\alpha_t = \frac{0.005}{\sqrt{1+t}}, \beta_t = \frac{0.01}{\sqrt{1+t}}, \gamma_t = \frac{0.1}{\sqrt{1+t}}, \sigma = 1, T = 10^6$. Note that multiplying small constants to these stepsizes does not affect our theoretical results.

For the zeroth-order method proposed in Fazel et al. (2018), we set $z = 5000, l = 20, r = 0.1$, stepsize $\eta = 0.01$ and iteration number $J = 1000$ for the first numerical example; while in the second example, we set $z = 20000, l = 50, r = 0.1, \eta = 0.01, J = 1000$. We choose different parameters based on the trade-off between better performance and fewer sample complexity.

For the double loop AC proposed in Yang et al. (2019), we set for both examples $\alpha_t = \frac{0.01}{\sqrt{1+t}}, \sigma = 0.2, \eta = 0.05$, inner-loop iteration number $T = 500000$ and outer-loop iteration number $J = 100$. We note that the algorithm is fragile and sensitive to the practical choice of these parameters. Moreover, we found that it is difficult for the algorithm to converge without an accurate critic estimation in the inner-loop. In our implementation, we have to set the inner-loop iteration number to $T = 500000$ to barely get the algorithm converge to the global optimum. This nevertheless demands a significant amount of computation. Higher $T$ iterations can yield more accurate critic estimation, and consequently more stable convergence, but at a price of even longer running time. We run the outer-loop for 100 times for each run of the algorithm. We run the whole algorithm 10 times independently to get the results shown in Figure. With parallel computing implementation, it takes more than 2 weeks on our desktop workstation (Intel Xeon(R) W-2225 CPU @ 4.10GHz × 8) to finish the computation. In comparison, it takes about 0.5 hour to run the single-sample single-timescale AC and 5 hours for the zeroth-order method.

---

**Algorithm 2** Zeroth-order Natural Policy Gradient

---

Input: stabilizing policy gain $K_0$ such that $\rho(A - BK_0) < 1$, number of trajectories $z$, roll-out length $l$, perturbation amplitude $r$, stepsize $\eta$
**while** updating current policy **do**
    **Gradient Estimation:**
    **for** $i = 1, \cdots, z$ **do**
        Sample $x_0$ from $\mathcal{D}$
        Simulate $K_j$ for $l$ steps starting from $x_0$ and observe $y_0, \cdots, y_{l-1}$ and $c_0, \cdots, c_{l-1}$.
        Draw $U_i$ uniformly over matrices such that $\|U_i\|_F = 1$, and generate a policy $K_{j,U_i} = K_j + rU_i$.
        Simulate $K_{j,U_i}$ for $l$ steps starting from $x_0$ and observe $c'_0, \cdots, c'_{l-1}$.
        Calculate empirical estimates:

$$\widehat{J^i_{K_j}} = \sum_{t=0}^{l-1} c_t, \ \widehat{\mathcal{L}^i_{K_j}} = \sum_{t=0}^{l-1} y_t y_t^\top, \ \widehat{J_{K_{j,U_i}}} = \sum_{t=0}^{l-1} c'_t.$$

    **end for**
    Return estimates:

$$\widehat{\nabla J(K_j)} = \frac{1}{z} \sum_{i=1}^{z} \frac{\widehat{J_{K_{j,U_i}}} - \widehat{J^i_{K_j}}}{r} U_i, \ \widehat{\mathcal{L}_{K_j}} = \frac{1}{z} \sum_{i=1}^{z} \widehat{\mathcal{L}^i_{K_j}}.$$

    **Policy Update:**
    $K_{j+1} = K_j - \eta \widehat{\nabla J(K_j)} \widehat{\mathcal{L}_{K_j}}^{-1}$.
    $j = j + 1$.
**end while**

---

---

**Algorithm 3** Double-loop Natural Actor-Critic

---

Input: Initial policy $\pi_{K_0}$ such that $\rho(A - BK_0) < 1$, stepsize $\gamma$ for policy update.
**while** updating current policy **do**
  **Gradient Estimation:**
  Initialize the primal and dual variables by $v_0 \in \mathcal{X}_\Theta$ and $\omega_0 \in \mathcal{X}_\Omega$, respectively.
  Sample the initial state $x_0 \in \mathbb{R}^d$ from stationary distribution $\rho_{K_j}$. Take action $u_0 \sim \pi_{K_j}(\cdot|x_0)$
  and obtain the reward $c_0$ and the next state $x_1$.
  **for** $i = 1, 2, \cdots, T$ **do**
    Take action $u_t$ according to policy $\pi_{K_j}$, observe the reward $c_t$ and the next state $x_{t+1}$.
    $\delta_t = v_{t-1}^1 - c_{t-1} + [\phi(x_{t-1}, u_{t-1}) - \phi(x_t, u_t)]^\top v_{t-1}^2$.
    $v_t^1 = v_{t-1}^1 - \alpha_t[\omega_{t-1}^1 + \phi(x_{t-1}, u_{t-1})^\top \omega_{t-1}^2]$.
    $v_t^2 = v_{t-1}^2 - \alpha_t[\phi(x_{t-1}, u_{t-1}) - \phi(x_t, u_t)] \cdot \phi(x_{t-1}, u_{t-1})^\top \omega_{t-1}^2$.
    $\omega_t^1 = (1 - \alpha_t)\omega_t^1 + \alpha_t(v_{t-1}^1 - c_{t-1})$.
    $\omega_t^2 = (1 - \alpha_t)\omega_t^2 + \alpha_t\delta_t\phi(x_{t-1}, u_{t-1})$.
    Project $v_t$ and $\omega_t$ to $v_0 \in \mathcal{X}_\Theta$ and $\omega_0 \in \mathcal{X}_\Omega$.
  **end for**
  Return estimates:
$$\widehat{v}^2 = (\sum_{t=1}^T \alpha_t v_t^2)/(\sum_{t=1}^T \alpha_t), \ \widehat{\Theta} = \text{smat}(\widehat{v}^2).$$

  **Policy Update:**
  $K_{j+1} = K_j - \eta(\widehat{\Theta}^{22} K_j - \widehat{\Theta}^{21})$.
  $j = j + 1$.
**end while**

---

