# OpenReview forum: "Finite-time Analysis of Single-timescale Actor-Critic on Linear Quadratic Regulator"
_ICLR.cc/2023/Conference — Submitted to ICLR 2023_

### Official Review · Reviewer_KvYy · 2022-10-24

**Confidence:** 4
**Correctness:** 3
**Technical Novelty And Significance:** 3
**Empirical Novelty And Significance:** Not applicable
**Recommendation:** 6

**Clarity, Quality, Novelty And Reproducibility:**

Overall, the paper is easy to follow. The quality and originality are fair-to-good.

**Strength And Weaknesses:**

Strength:
- The work considers the single-timescale update, which is more practical than the double-loop update analyzed in previous works.
- The paper proposes a new analysis framework for the algorithm and establishes the sample complexity of $\tilde {\mathcal{O}}(\epsilon^{-2})$, which matches the best existing sample complexity for single-timescale AC algorithm.

Weakness:
- Previous works [1, 2] which study the LQR problem seems to have weaker assumption: They only require that initial policy is stable, instead of being stable for each iteration, as assumed in this paper.
- Since the paper only presents the convergence of the natural AC instead of the AC for the LQR, I suggest the author to add more discussions on the convergence of AC in the introduction.



**Summary Of The Paper:**

The work considers the linear quadratic regulator problem, and studies the natural actor critic algorithm’s convergence under the single-timescale update, where the step-size of actor and critic variables are proportional by a constant. They show that the algorithm attains the global optimum with the sample complexity of $\tilde {\mathcal{O}}(\epsilon^{-2})$.

**Summary Of The Review:**

I have the following main comments:

- The title and the abstract are quite misleading, in which the authors claim that they obtain the convergence of single-timescale AC under LQR problem. After going through the paper, the authors only consider natural AC (NAC). AC and NAC are two quite different algorithms. Hence, the authors must revise any related claims accordingly to reduce confusion.

- From my understanding, Assumption 4.2 is mainly used to ensure the existence of stationary distribution. Is it possible to remove the assumption by considering using the sample from a trajectory? (as in [1, 2])

- Can you also provide convergence of AC under the online update? Even without the knowledge of the model? NAC often has more complex oracle than AC. What is the motivation to consider NAC rather than AC?

- There are works [3, 4, 5] that study the single-timescale AC’s sample complexity for finding a stationary point under the general cost function (rather than LQR). Can their results be extended to LQR problem and obtain the same rate for global convergence? In LQR, the gradient domination property (also known as Polyak-Łojasiewicz inequality) implies that the optimal gap converges in the sample rate as the gradient norm square.

- The authors missed the related reference [4], which gives convergence results for single-timescale AC under the general cost function.


[1] Global Convergence of Policy Gradient Methods for the Linear Quadratic Regulator.

[2] On the Global Convergence of Actor-Critic: A Case for Linear Quadratic Regulator with Ergodic Cost.

[3] Closing the gap: Tighter analysis of alternating stochastic gradient methods for bilevel problems.

[4] Finite-Time Analysis of Fully Decentralized Single-Timescale Actor-Critic.

[5] A small gain analysis of single timescale actor critic.

---

> ### Author Response · Authors · 2022-11-09
> **Thanks for your comments!**
>
> **C1 (on AC and NAC):**
>
> Thanks for your advice. The reason why we consider NAC instead of AC is that the natural gradient of LQR has a simple form, which eliminates the burden of estimating the covariance matrix $D_K$. It is standard to apply NAC for solving LQR as done in [2]. For clarity, we will revise single-timescale AC to single-timescale NAC.
>
> **C2(sample from a trajectory):**
>
> Yes, it could be possible. However, our aim is to test whether the more practical single-sample single-timescale NAC can solve the LQR. We found that this widely used algorithm can indeed solve the LQR optimally and it enjoys much superior sample efficiency of $\mathcal{O}(\epsilon^{-2})$ compared to $\mathcal{O}(\epsilon^{-4})$ obtained in [1] and $\mathcal{O}(\epsilon^{-5})$ in [2]. This superiority is also empirically validated by our numerical experiments where we compared the single-sample single-timescale NAC with [1] and [2], which justifies the practical wisdom of single-sample single-timescale NAC algorithms. The analysis of Markovian samples from a single trajectory will be our future work.
>
> **C3(on convergence of AC):**
>
> It is challenging.
> * To our best endeavors, it is possible to prove the finite convergence rate for AC with online updates only under finite action space. Nevertheless, the most challenging part in LQR is that the state-action space is continuous which leads to $|\mathcal{A}|=\infty$. Most previous online analyses will fail under infinite action space, which makes it significantly more challenging for the analysis of the online sampling case.
> * As mentioned earlier, the motivation for considering NAC is that it has a simpler form for solving LQR than  AC. Analyzing the convergence of AC in LQR can be significantly more difficult.
>
> **C4(comparison with [3,4,5]):**
>
>
> To our best efforts, all the results of [3,4,5] cannot be extended to the LQR setting easily. Particularly, [4,5] considered the tabular setting where both state space and action space are finite. In their proof, they use total state numbers ($|\mathcal{S}|$) and total action numbers ($|\mathcal{A}|$) as upper bounds extensively. In LQR setting, both $|\mathcal{S}|$ and $|\mathcal{A}|$ are infinity. In [3], they also considered finite action space. Their Proposition 7 will not hold for the LQR case.
>
> **C5(on reference [4]):**
>
> Similar to the above comparison, their results cannot apply to LQR. We would like to highlight that the LQR is more challenging due to its nature of unbounded continuous state-action space and unbounded reward. Nevertheless, we thank the reviewer for directing us to the relevant work for a more complete literature review.

---

### Official Review · Reviewer_Me62 · 2022-10-24

**Confidence:** 4
**Correctness:** 4
**Technical Novelty And Significance:** 2
**Empirical Novelty And Significance:** 2
**Recommendation:** 5

**Clarity, Quality, Novelty And Reproducibility:**

In the third paragraph of Section 1, the explanation about single-timescale AC is similar to that about two-timescale AC. The statement should include the difference between the single-timescale and two-timescale AC. Further, there are multiple typos and grammatical errors.

**Strength And Weaknesses:**

For the policy defined in (7), authors can provide better exploration methods such as optimism or Thompson sampling. The suggested exploration methods can over-explore over time. Also, policies with better exploration methods may improve the theoretical results.

For experiments, it would be interesting to see comparisons between the single-timescale and two-timescale AC as well.


**Summary Of The Paper:**

This paper proposes a single-sample single-timescale Actor-Critic method for the linear quadratic regulator (LQR). The main contribution with respect to the available literature is that the authors suggest the epsilon-optimal solution with a sample complexity \tilde{O}(epsilon^2) and provide some convergence analyses.

**Summary Of The Review:**

Post rebuttal:

I found the paper unready for publication. The importance and novelty of the results is unclear, e.g., the superiority of the suggested method over the existing ones that are faster, more flexible, and can be implemented in an online manner. The technical assumptions, e.g. stability, are strong, while they are studied in the literature. The literature review is also incomplete in other places, e.g. RL in LQR. The main theoretical contribution lacks scalability and it is not clear how the failure probability affects the convergence results. Also on the main theorem, the main convergence rate which is the last one, is not as strong as one expects, and I believe that it can be improved to 1/T.

---

> ### Author Response · Authors · 2022-11-09
> **Thanks for your comments!**
>
> **C1 (on exploration):**
>
> Thanks for your advice. In our work, we add Gaussian noise to encourage exploration so that we can solve LQR optimally. It is possible to consider more advanced exploration strategies. But it generally requires a very different analysis. Major work is needed that is beyond the scope of the current paper. Besides, to the best of our knowledge, even for the tabular Q-learning with upper confidence bound exploration, the existing best theoretical analysis [1] still has not shown improved performance over general random sampling. The analysis for more complicated exploration strategies remains very challenging and requires more advanced technicalities.
>
> We thank the reviewer for pointing out the interesting direction and we will leave the exploration for our future work.
>
> **C2 (comparison between single-timescale and two-timescale AC):**
>
> Thanks for your advice. We have added the comparison in our Figure 1.(b). We set the timescales for actor and critic are $\mathcal{O}((1+t)^{-0.6})$ and $\mathcal{O}((1+t)^{-0.4})$ respectively since such a pair of timescales has best performance according to [2]. It shows that single-sample two-timescale actor-critic can also outperform zeroth order method and double-loop actor-critic due to its single-loop nature. However, single-timescale AC is superior than two-timescale AC which is consistent with the theoretical findings in our paper and [2] where they show that two-timescale AC enjoys a $\mathcal{O}(\epsilon^{-2.5})$ sample complexity while ours are $\mathcal{O} (\epsilon^{-2})$.
>
> [1] Q-learning with UCB Exploration is Sample Efficient for Infinite-Horizon MDP
>
> [2] A Finite-Time Analysis of Two Time-Scale Actor-Critic Methods

---

> ### Author Response · Authors · 2022-12-02
> **Further Response**
>
> We hope our first round of responses has addressed the reviewer's concerns satisfactorily.
>
> Regarding the reviewer's new comments, we would like to emphasize that for single-sample single-time actor-critic, the gradient estimation and the value estimation are much noisier than multi-sample actor-critic algorithms. It may be possible for the latter to achieve a better rate of 1/T. But there will be a big gap for the former. To our knowledge, the results in this paper are the best available so far.
>
> In addition, our focus is on the actor-critic algorithm. We are afraid that the RL in LQR literature would be too broad and out of the scope of our review.
>
> The other newly added comments seem to follow from Reviewer HWQj. Please refer to our response there for further clarification.

---

### Official Review · Reviewer_HWQj · 2022-10-25

**Confidence:** 4
**Correctness:** 2
**Technical Novelty And Significance:** 2
**Empirical Novelty And Significance:** Not applicable
**Recommendation:** 3

**Clarity, Quality, Novelty And Reproducibility:**

Overall the main paper is easy to follow. The proof seems a bit convoluted but I think this is hard to avoid due to the theoretical nature of this work.

**Strength And Weaknesses:**

Major Comments:

(1) The claim of "single time-scale" is not correct. The constant $c_\alpha$ in the stepsize $\alpha_t$ is in fact not a constant, but depends on the final iteration number $T$, as shown in Eq. (52). Since $t\leq T$, we still have $\alpha_t/\beta_t\rightarrow 0$ as $t$ goes to infinity, and the algorithm is in fact two time-scale. In view of this, the contribution of this paper is unclear.

(2) As a follow-up comment to (1), the motivation of using diminishing step sizes is that the agent does not need to choose the stepsize based on the final iteration number $T$, otherwise one can simply use constant stepsize designed based on $T$. Since $c_\alpha$ depends on $T$, this contradicts to the motivation of using diminishing stepsizes.

(3) The authors tried to justify the i.i.d. sampling assumption by stating that one can wait until the Markov chain is sufficiently mixed and then collect one sample. However, this contradicts to the "single sample" claim made in this paper. Also, in terms of analysis, the i.i.d. assumption greatly simplifies the proof, and hence is not mild in this viewpoint. In addition, there are many existing papers successfully handled Markovian sampling.

(4) While the authors justified the use of Assumption 4.1 and Assumption 4.2 by citing existing papers and providing numerical simulations, these two assumptions are not logical. Before we run the algorithm, there is no guarantee on the size of $K_t$ and the stability of the system. Assuming something that may or may not happen in the future does not make sense.

(5) When talking about finite-time bound, one should avoid using the big O notation and explicitly characterize all the constants. Theorems involving big O notation are not finite-time bounds but asymptotic bounds.

Minor Comments:

(1) Please change the title to "... natural actor-critic ... " to avoid confusion with vanilla actor-critic (without fisher information preconditioner).

(2) There are many related papers studying natural actor-critic (even beyond LQR) and established $O(1/\epsilon^2)$ sample complexity, just to list a few:

[1] Xiao, L. (2022). On the convergence rates of policy gradient methods. arXiv preprint arXiv:2201.07443.

[2] Lan, G. (2022). Policy mirror descent for reinforcement learning: Linear convergence, new sampling complexity, and generalized problem classes. Mathematical programming, 1-48.

[3] Chen, Z., & Maguluri, S. T. (2022, May). Sample Complexity of Policy-Based Methods under Off-Policy Sampling and Linear Function Approximation. In International Conference on Artificial Intelligence and Statistics (pp. 11195-11214). PMLR.

(3) The authors should make clear about what is established in the literature and what is the contribution of this work. For example, in Section 3, is everything before the algorithm a contribution of this work or already established in literature (e.g., Proposition 3.1)?

(4) Notation such as svec and smat should be properly defined.



**Summary Of The Paper:**

This paper proposes a single time-scale actor-critic (AC) algorithm to solve the LQR problem. The authors establish an $O(1/\sqrt{T})$ rate of global convergence. Numerical simulations are provided to demonstrate the performance of the proposed algorithm.

**Summary Of The Review:**

Since the proposed algorithm is in fact two time-scale, and there are many existing papers analyzing two time-scale actor-critic and its variants, the main contribution (i.e., analysis of single time-scale actor-critic) claimed by the authors is unclear. In addition, some of the assumptions are not realistic. In view of these two point, I cannot recommend accepting this paper to ICLR.

---

> ### Author Response · Authors · 2022-11-09
> **Thanks for reviewing our paper! (Response--Part II)**
>
> **C5 (on the convention of finite-time expression):**
>
> * We have explicitly characterized all the necessary constants in the proofs before the last step of the analysis of the interconnected system. But we believe this helps to better convey our proof ideas, one of our main contributions. One can easily keep all the constants in the interconnected system analysis and get the order for all constants.
> * In addition, we didn't refer to the main results of our theorem as the finite-time bound. To be more focused on the key factors and for ease of comprehension, We chose to only show the convergence rate in terms of the iteration number parameter $T$.
>
> **C6 (on the contribution and the provided references):**
>
> Thanks for seeking clarification. Our main contribution is that we for the first time show that this practice single-sample single-timescale natural actor-critic can solve the LQR problem optimally (please also refer to the 3 bullet points that summarize our contribution in the Introduction). In particular, all the theorems and propositions are established by us, including Proposition 3.1. Our work stands at the cutting edge of understanding the behavior and limits of the practical actor-critic algorithms on continuous state-action space problems. These problems are the most practically useful cases.
>
> For continuous state-action space, the cardinalities of the states ($|\mathcal{S}|$) and actions ($|\mathcal{A}|$) are infinity which makes most prior results inapplicable, since they heavily rely on the boundedness of $|\mathcal{S}|$ and $|\mathcal{A}|$. In fact, all the references provided by the reviewer neither consider the infinite state-action space (in fact, they all deal with simple tabular cases) nor consider the single-timescale actor-critic. These methods cannot be applied to analyze the challenging LQR case in this paper.
>
> **C7 (definitions of svec and smat):**
>
> We have added references/definitions to the two notations in Appendix for clarity.
>
> Overall, we thank you for your review and comments. Hope our responses have addressed all your concerns satisfactorily. Feel free to leave further comments if there is still anything that is unclear.

---

> > ### Comment · Reviewer_HWQj · 2022-11-24
> > **After Response**
> >
> > Thank the authors for their detailed response. After reading the response, some of comments are addressed but still there are some major concerns. Therefore, I will maintain my score.
> >
> > C1: Proposition A.4. seems to be critical for the analysis. In particular, the fact that the stepsizes are single time-scale relies on the boundedness of $\bar{U}$ stated in Proposition A.4. However, Proposition A.4. is true under Assumption 4.1 and Assumption 4.2, both of which are not logical. If the iterates are not uniformly bounded and uniformly stable as assumed, the single time-scale analysis breaks down.
> >
> > It may be a better idea to state the results as: with probably $1-\delta$, we have $J(K_t)-J(K^*) \leq$ function of $T$ and $\delta$, instead choosing $\delta$ to be a numerical constant. A main advantage of deriving high probability bounds is that we can see explicitly how the tail behaves, i.e., exponential tail $\log(1/\delta)$ or polynomial tail $1/\delta$. This is not clear in this paper due to choosing $\delta$ as a numerical constant and the big O notation.
> >
> > C3: While it is not completely satisfactory, I agree with the authors that the i.i.d. assumption is not a deal breaker, and it is a valid first-step in theoretically analyzing a stochastic iterative algorithm.
> >
> > C4: I do not agree with the authors that this assumption is valid simply because it seems true empirically and was used in existing literature. Since this paper is theoretical in its nature, this assumption is indeed a hole and should be fixed. There are already existing papers that try to remove this assumption by modifying the algorithm design, such as truncating the estimate of the value function.
> >
> > C6: If I understand correctly, the main challenge here is that the state-action space is continuous. I believe having continuously infinite state-action space alone is not a major difficulty, the major difficulty here is that the reward can be unbounded because $|S||A|=\infty$. However, this difficulty is overcome by imposing Assumption 4.1 and Assumption 4.2.

---

> > > ### Author Response · Authors · 2022-12-02
> > > **Further Response**
> > >
> > > **On unbounded reward and main challenge**:
> > >
> > > The unboundedness of the reward is incurred by the unbounded state-action pair since the reward is a quadratic form of state-action pair. We address this difficulty by giving state-action pairs an upper bound which holds with high probability.
> > >
> > > Since the finite-time analysis for single-sample single-timescale algorithms are not well established, the continuous state-action setting makes it more challenging because all prior results are inapplicable. We are considering a setting that no one tackled before.
> > >
> > > **On the assumptions**:
> > >
> > > We notice that our major disagreement lies in the two assumptions. We justify that the lack of stability guarantee is a common problem in single-sample single-loop (natural) actor-critic algorithms. We emphasis that the lack of stability is not due to the deficit of our analysis method, but due to the performance limit of this kind of algorithms. Actually, assuring stability is not difficult as long as one change to the double-loop algorithm (Yang et al. (2019)) or multi-sample algorithm (Krauth, Tu, and Recht (2019)) since both structures can solve the policy evaluation problem completely. However, we never consider to change the algorithm because our starting point is to test the performance of this classic algorithm under continuous control task. Therefore, there seems a mismatch of our focus.
> > >
> > > Overall, we thank the reviewer for reviewing our manuscript and we understand that everyone has different research tastes.

---

> ### Author Response · Authors · 2022-11-09
> **Thanks for reviewing our paper! (Response--Part I)**
>
> Thanks for the comments. We are afraid that the reviewer significantly misunderstood the focus, technicalities, and contribution of our paper. In particular, our paper focuses on the finite time analysis of the most commonly used single-timescale natural AC (actor-critic) when applied to LQR (linear quadratic regulator), rather than proposing a new algorithm for solving LQR. Most of the existing works (including those very recent ones pointed out by the reviewer) can only analyze the simple tabular (finite state-action space) AC. There is a huge gap in analysis when moving from tabular to continuous state-space cases. As a case study, analyzing LQR serves as the first step toward analyzing the more general continuous state-action space problems.
> We further address the reviewer's concerns in detail below.
>
> **C1 (on the single-timescale stepsize):**
>
> We would like to clarify that the considered setting is indeed "single-timescale".
>
> * The $\log(T)$ term in the threshold of $c_{\alpha}$ (equation (52)) is only introduced to have an asymptotic high probability of $1-2T^{-10}$ in the main theorem. That is, the derived convergence rate provably holds with a higher probability at a smaller stepsize ratio (only their ratio, not the absolute value of the stepsizes, matters). We could easily present the result with a fixed high probability of $1-\delta$, as many other papers did, in which case, the threshold will be a constant independent of $T$. In the updated manuscript, to avoid any confusion, we fixed the high probability to be $1-10^{-10}$ (which is easily achieved by using the same concentration inequality). In the updated manuscript, we highlight the revision in red for your ease of reference (mainly in Proposition A.3 and its proof).
> * ``two-timescale" stepsize is considered mainly for the easiness of establishing convergence, in which the analysis heavily relies on the nice property of $\lim\limits_{t\to\infty}\frac{\alpha_t}{\beta_t}=0$. Unlike the "two-timescale", the single-timescale stepsize considered in our paper does not have such a property and our proof does not require such a property either. Once the desired high probability is chosen, the constant stepsize ratio can be determined, which is independent of the total iteration number $T$, and our convergence rate hold for all $T$.
>
> **C2 (on the diminishing stepsize):**
>
> As explained in C1, our stepsize choice is essentially independent of $T$. Only the ratio of the stepsizes depends on the desired high probability. In particular, we provide a fixed high probability $1-10^{-10}$ in the updated manuscript so that the stepsizes is independent of $T$.
>
> **C3 (on the stationary distribution sampling):**
>
> * By single sample, we mean (also see the paragraph under Algorithm 1 in the manuscript) only one sample is used to update the critic per actor
> step. Our analysis is based on such a single-sample setting. Hence, we think the claim of "single sample" is clearly justified, although running the simulator many steps but only sampling the state and reward of the last step might not be efficient in practice.
> * Note that many existing theoretical works start with i.i.d samples from the stationary distribution. It is widely recognized as the first important step toward the analysis of more practical algorithms.
> * The "mild requirement" in the same paragraph below Algorithm 1 mainly refers to the practical side---the fact that sampling from stationary distribution can be achieved relatively easily in practice if the Markov chain is geometrically mixing. Because one doesn't have to wait for too long to sample from the simulator. This is one point of confusion due to our expression. We have updated the paper accordingly to be more accurate.
> * Theoretical-wise, we did acknowledge that i.i.d sampling from stationary distribution simplifies the analysis. Existing works that consider Markovian sampling are all under the finite action space ($|\mathcal{A}|<+\infty$). Their methods cannot be easily applied to analyzing LQR which is with unbounded continuous action space ($|\mathcal{A}|=+\infty$). In fact, the LQR case is significantly more challenging. Our preliminary research indicates that the proof hinges on the characterization of the Lipschitz property of the total variation norm between two multi-variate Gaussian distributions, which is highly non-trivial.
>
> **C4 (on the two assumptions):**
>
> We would not say that these assumptions "do not make sense", especially when we can easily show many examples that satisfy these assumptions. In fact, a large body of recent and earlier works assume similar ("may or may not happen in the future") conditions to facilitate the analysis. These assumptions indicate the key challenges of the considered problems from a different angle. Finding easier-to-check conditions that guarantee boundedness and uniform stability will be our future work.

---

### Author Response · Authors · 2022-11-09
**General Response to All Reviewers**

The authors thank all the reviewers for their time and efforts in reviewing our manuscript. We would like to emphasize that the most challenging and practically useful applications are typically those with continuous state-action space. The references provided by the reviewers nevertheless can only consider tabular problems (i.e., with finite state-action space). These works are inapplicable to the more challenging LQR case since they heavily rely on the boundedness of $|\mathcal{S}|$ and $|\mathcal{A}|$, which for LQR are infinite. In fact, there is a huge gap in the analysis when moving from tabular to continuous state-action space cases. Our work stands at the cutting edge of understanding the behavior and limits of the practical actor-critic algorithms for solving problems with continuous state-action space.

---

### Comment · Area_Chair_w3iM · 2022-11-09
**Author's Response posted.**

Dear Reviewers,

The authors have posted their responses to each review. Please take a look at the response and see if it addresses your concerns. If there are further clarifications/revisions that you would like the authors to make, please respond to the respective comments. Ideally, we would like to have several rounds of conversation between the reviewers and the authors to ensure that opinions on both sides are well-elaborated.

Thank you very much for your work,
Your AC

---

### Decision · Program_Chairs · 2023-01-20

**Decision:**

Reject

**Justification For Why Not Higher Score:**

NA

**Justification For Why Not Lower Score:**

NA

**Metareview: Summary, Strengths And Weaknesses:**

This paper makes an attempt to provide an analysis for a single-timescale Natural Actor-Critic (NAC) algorithm on Linear Quadratic Regulator (LQR). The theoretical problem is well-motivated. However, it seems that the main drawback of the current analysis is its dependence on a strong algorithm-dependent stability assumption, thus falling into the "chicken and egg" scenario. I believe this is a severe problem and requires significant effort to address. Therefore, I must recommend rejection at this point.